# Exponential Generalization Bounds with Near-Optimal Rates for $L_q$-Stable Algorithms

**Xiao-Tong Yuan**
School of Intelligence Science and Technology
Nanjing University, Suzhou, 215163, China
`xtyuan1980@gmail.com`

**Ping Li**
LinkedIn Ads
700 Bellevue Way NE, Bellevue, WA 98004, USA
`pinli@linkedin.com`

## Abstract

The *stability* of learning algorithms to changes in the training sample has been actively studied as a powerful proxy for reasoning about generalization. Recently, exponential generalization and excess risk bounds with near-optimal rates have been obtained under the stringent and distribution-free notion of uniform stability (Bousquet et al., 2020; Klochkov & Zhivotovskiy, 2021). In the meanwhile, under the notion of $L_q$-stability, which is weaker and distribution dependent, exponential generalization bounds are also available yet so far only with sub-optimal rates. Therefore, a fundamental question we would like to address in this paper is whether it is possible to derive near-optimal exponential generalization bounds for $L_q$-stable learning algorithms. As the core contribution of the present work, we give an affirmative answer to this question by developing strict analogues of the near-optimal generalization and risk bounds of uniformly stable algorithms for $L_q$-stable algorithms. Further, we demonstrate the power of our improved $L_q$-stability and generalization theory by applying it to derive strong sparse excess risk bounds, under mild conditions, for computationally tractable sparsity estimation algorithms such as Iterative Hard Thresholding (IHT).

## 1 Introduction

A fundamental issue in statistical learning is to bound the generalization error of a learning algorithm for understanding its prediction performance on unseen data. It has long been recognized in literature that one of the key characteristics that permits learning algorithms to generalize is the *stability* of estimated model to perturbations in training data. The idea of using algorithmic stability as a proxy for generalization performance analysis dates back to the seventies (Rogers & Wagner, 1978; Devroye & Wagner, 1979). Since the seminal work of Bousquet & Elisseeff (2002), the search for generalization bounds under various notions of algorithmic stability has been a flourishing area of learning theory (Zhang, 2003; Mukherjee et al., 2006; Shalev-Shwartz et al., 2010; Kale et al., 2011; Hardt et al., 2016; Celisse & Guedj, 2016; Bousquet et al., 2020).

As one may expect, the stronger an algorithmic stability criterion is, the sharper the resulting generalization bound will be. On one end, exponential generalization bounds can be guaranteed by approaches under the most stringent notion of uniform stability (Bousquet & Elisseeff, 2002; Bousquet et al., 2020), which requires the change in the prediction loss to be uniformly small regardless data distribution. Despite the strength of generalization, the distribution-free nature makes uniform stability too restrictive to be fulfilled, e.g., by learning rules with unbounded losses (Celisse & Guedj, 2016). On the other end, based on some weaker and distribution dependent notions of stability such as hypothesis stability and mean-square stability, only polynomial generalization bounds seem possible in general cases, although the corresponding stability criteria are more amenable to verification (Bousquet & Elisseeff, 2002). These observations have prompted the development of $L_q$-stability, as an in-between state, to achieve the best of two worlds (Celisse & Guedj, 2016; Abou-Moustafa & Szepesvári, 2019): it generalizes the notion of hypothesis stability from $\ell_1$-norm criterion to $L_q$-norm criterion for $q \geq 2$ but remains distribution dependent and thus is weaker than uniform stability; in the meanwhile it can still achieve similar exponential generalization bounds to those of uniformly stable algorithms (Bousquet & Elisseeff, 2002).

By far, the best known (and near-optimal) rates about exponential generalization bounds are offered by approaches based on uniform stability and certain fine-grained concentration inequalities for sum of functions of independent random variables (Feldman & Vondrák, 2019; Bousquet et al., 2020). These rates are substantially sharper than those of Bousquet & Elisseeff (2002), which are implied by a naive application of McDiarmid's inequality, in terms of the overhead factors on stability coefficients. While it has long been known that the low probability of failure (over sample) can be handled via developing modified bounded-difference inequalities (Rakhlin et al., 2005), it still remains less clear how to simply adapt these existing techniques to the more sophisticated frameworks of Feldman & Vondrák (2019); Bousquet et al. (2020) to obtain sharper exponential bounds. Particularly for $L_q$-stable learning algorithms, the state-of-the-art exponential generalization bounds are derived based on the moments or exponential extensions of the Efron-Stein inequality (Celisse & Guedj, 2016; Abou-Moustafa & Szepesvári, 2019), which yield similar rates of convergence to those of Bousquet & Elisseeff (2002) and thus are suspected to be *sub-optimal*.

Given the above observed gap in rates of convergence between the generalization bounds under uniform stability and $L_q$-stability, the following question is naturally raised:

> *Is it possible to derive sharper exponential generalization bounds for $L_q$-stable learning algorithms that match those recent breakthrough results for uniformly stable algorithms?*

As the core contribution of the present work, we give an affirmative answer to this open question by developing strict analogues of the near-optimal generalization bounds of uniformly stable algorithms for $L_q$-stable algorithms. The main results of our work confirm that the notion of $L_q$-stability serves as a neat yet powerful tool for extending those best-known generalization bounds to a broad class of non-uniformly stable algorithms. To illustrate the importance of our theory, we have applied the improved analysis of $L_q$-stable algorithms to derive sharper exponential risk bounds for computationally tractable sparsity recovery estimators, such as the Iterative Hard Thresholding (IHT) algorithms widely used in high dimensional sparse learning (Blumensath & Davies, 2009; Foucart, 2011; Jain et al., 2014). This application also serves as a main motivation of our study.

**Notation.** Here we provide some notation that will be frequently used throughout the paper. Let $S = \{Z_1, Z_2, ..., Z_N\}$ be a set of independent random data samples valued in some measurable set $\mathcal{Z}$. For any indices set $I \subseteq [N] := \{1, ..., N\}$, we denote by $S_I = \{Z_i, i \in I\}$ and $S_{\bar{I}} = S \setminus S_I$. We denote by $S' = \{Z'_1, Z'_2, ..., Z'_N\}$ another i.i.d. sample from the same distribution as that of $S$ and we write $S^{(i)} = \{Z_1, ..., Z_{i-1}, Z'_i, Z_{i+1}, ..., Z_N\}$. For a real-valued random variable $Y$, its $L_q$-norm for $q \geq 1$ is defined by $\|Y\|_q = (\mathbb{E}[|Y|^q])^{1/q}$. By definition it can be verified that $\forall q \geq 2$,

$$\|Y\|_q^2 = (\mathbb{E}[|Y|^q])^{2/q} = \left(\mathbb{E}[|Y^2|^{q/2}]\right)^{2/q} = \|Y^2\|_{q/2}. \tag{1}$$

Let $g : \mathcal{Z}^N \mapsto \mathbb{R}$ be some measurable function and consider the random variable $g(S) = g(Z_1, ...Z_N)$. For $g(S)$ and any index set $I \subseteq [N]$, we define the following abbreviations:

$$g(S_I) := \mathbb{E}[g(S) \mid S_I], \quad \|g\|_q(S_I) := (\mathbb{E}[|g(S)|^q \mid S_I])^{1/q}.$$

We say a real-valued function $f$ is $G$-Lipschitz continuous over the domain $\mathcal{W}$ if

$$|f(w) - f(w')| \leq G\|w - w'\|, \quad \forall w, w' \in \mathcal{W}.$$

For a pair of functions $f, g \geq 0$, we use $f \lesssim g$ (or $g \gtrsim f$) to denote $f \leq cg$ for some constant $c > 0$. We denote by $\text{supp}(w)$ the *support* of a vector $w$ which is the index set of non-zero entries of $w$.

## 1.1 SETUP AND PRIOR RESULTS

**Problem setup.** We consider a statistical learning algorithm $A : \mathcal{Z}^N \mapsto \mathcal{W}$ that maps a training data set $S$ to a model $A(S)$ in a closed subset $\mathcal{W}$ of an Euclidean space. The population risk and corresponding empirical risk evaluated at $A(S)$ are respectively given by

$$R(A(S)) := \mathbb{E}_Z[\ell(A(S); Z)] \text{ and } R_S(A(S)) := \frac{1}{N}\sum_{i=1}^{N} \ell(A(S); Z_i),$$

where $\ell : \mathcal{W} \times \mathcal{Z} \mapsto \mathbb{R}^+$ is a non-negative and potentially unbounded loss function whose value $\ell(w; z)$ measures the loss evaluated at $z$ with parameter $w$. As a classic fundamental issue in statistical learning, we are interested in deriving the upper bounds on the difference between population

and empirical risks, i.e., $|R(A(S)) - R_S(A(S))|$, which quantifies the generalization error of $A$. Let $R^* := \min_{w \in \mathcal{W}} R(w)$ be the optimal value of the population risk. We will also study how to upper bound $R(A(S)) - R^*$ (a.k.a. excess risk) which is of particular interest for understanding the population risk minimization performance of $A$.

We first introduce the concept of uniform stability (Bousquet & Elisseeff, 2002) which requires the change in the prediction loss to be uniformly small regardless the distribution of data.

**Definition 1** (Uniform stability). *A learning algorithm $A$ is said to have uniform stability with parameter $\gamma_u > 0$ if it satisfies the following uniform bound:*

$$\sup_{S, S^{(i)}, Z \in \mathcal{Z}} |\ell(A(S); Z) - \ell(A(S^{(i)}); Z)| \leq \gamma_u, \quad \forall i \in [N].$$

Given that the loss function $\ell$ is almost surely bound by $M$, Bousquet & Elisseeff (2002) showed that a large class of regularized empirical risk minimization (ERM) algorithms has uniform stability, and using McDiarmid's inequality yields the following exponential tail generalization bound that holds with probability at least $1 - \delta$ over the draw of $S$ for any $\delta \in (0, 1)$:

$$|R(A(S)) - R_S(A(S))| \lesssim \gamma_u \sqrt{N \log\left(\frac{1}{\delta}\right)} + M\sqrt{\frac{\log(1/\delta)}{N}}. \tag{2}$$

Recently, equipped with a strong concentration inequality for sums of random functions, Bousquet et al. (2020) established the following moments bound of uniformly stable algorithms for all $q \geq 2$:

$$\|R(A(S)) - R_S(A(S))\|_q \lesssim q\gamma_u \log(N) + M\sqrt{\frac{q}{N}}. \tag{3}$$

In view of the equivalence between tails and moments (see, e.g., Bousquet et al., 2020, Lemma 1), the above $L_q$-norm bound implies that for any $\delta \in (0, 1)$, the following tail bound holds with probability at least $1 - \delta$ over the draw of $S$:

$$|R(A(S)) - R_S(A(S))| \lesssim \gamma_u \log(N) \log\left(\frac{1}{\delta}\right) + M\sqrt{\frac{\log(1/\delta)}{N}}. \tag{4}$$

This bound substantially improves the classic result in Eq. (2) by reducing the overhead factor on stability coefficient from $\mathcal{O}\big(\sqrt{N \log(\frac{1}{\delta})}\big)$ to $\mathcal{O}(\log(N) \log(\frac{1}{\delta}))$. For example, in regimes such as regularized ERM where $\gamma_u \lesssim \frac{1}{\sqrt{N}}$ is usually the case, the convergence rate in Eq. (2) becomes vacuous as it is not vanishing in sample size, while the bound in Eq. (4) still guarantees $\mathcal{O}\big(\frac{\log(N) \log(\frac{1}{\delta})}{\sqrt{N}}\big)$ rate of convergence. Indeed, up to logarithmic factors on sample size and tail bounds, the rate in Eq. (4) is nearly optimal in the sense of a lower bound on sums of random functions by Bousquet et al. (2020). The bound in Eq. (4) can be extended to stochastic learning algorithms when the uniform stability (over data) holds with high probability over the internal randomness of algorithm (Feldman & Vondrák, 2019; Bassily et al., 2020). Under the generalized Bernstein condition (Koltchinskii, 2006) and based on the sharp concentration inequality for sums of random functions by Bousquet et al. (2020), Klochkov & Zhivotovskiy (2021) alternatively established the following deviation optimal excess risk bound that holds with probability at least $1 - \delta$ over the draw of $S$:

$$R(A(S)) - R^* \lesssim \Delta_{\text{opt}} + \mathbb{E}[\Delta_{\text{opt}}] + \gamma_u \log(N) \log\left(\frac{1}{\delta}\right) + \frac{(M + B) \log(1/\delta)}{N}, \tag{5}$$

where $\Delta_{\text{opt}} := R_S(A(S)) - \min_{w \in \mathcal{W}} R_S(w)$ represents the empirical risk sub-optimality of the algorithm on training data, and $B$ is the Bernstein condition constant as defined in Assumption 1.

While implying strong generalization guarantees, the uniform stability is also most stringent in the sense that it is distribution independent and hard to be fulfilled, e.g., by learning rules with unbounded losses. To address such an unpleasant restrictiveness, the notion of $L_q$-stability was alternatively introduced by Celisse & Guedj (2016) as a relaxation of uniform stability.

**Definition 2** ($L_q$-Stability). *For $q \geq 1$, a learning algorithm $A$ is said to have $L_q$-stability with parameter $\gamma_q > 0$ if it satisfies the following moment bound:*

$$\left\|\ell(A(S); Z) - \ell(A(S^{(i)}); Z)\right\|_q \leq \gamma_q, \quad \forall i \in [N].$$

In the above definition, the expectation associated with $L_q$-norm is taken over $S, S^{(i)}, Z$, and the internal random bits of $A$, if any (such as in the case of stochastic learning algorithms). Note that slightly different from that of Celisse & Guedj (2016), the random variable $Z$ in the above definition is not necessarily required to be independent of $S$ and $S^{(i)}$. By definition, $L_q$-stability is distribution dependent and thus is weaker than uniform stability which can be regarded as a special case of $L_q$-stability with $\gamma_q \equiv \gamma$ for some $\gamma > 0$. Particularly for $q = 1$ and $q = 2$, the $L_q$-stability reduces to the notions of hypothesis stability (Bousquet & Elisseeff, 2002) and mean-square stability (Kale et al., 2011), respectively. For an instance, it has been shown that the classical ridge regression model with unbounded responses has $L_q$-stability for all $q \geq 1$ rather than uniform stability (Celisse & Guedj, 2016). As a novel and concrete example, we will see shortly in Section 3 that $L_q$-stability plays a crucial role for deriving strong sparse excess risk bounds for sparsity estimation algorithms such as IHT (Jain et al., 2014; Yuan et al., 2018). Alternatively, the definition of $L_q$-stability can be extended to the $L_q$-argument-stability as $\left\| \|A(S) - A(S^{(i)})\| \right\|_q \leq \gamma_q$, which generalizes the concept of uniform argument stability (Bassily et al., 2020) to the $L_q$-norm criterion. Obviously $L_q$-argument-stability is not at all relying on the random argument $Z$ and it implies $L_q$-stability for Lipschitz losses.

The following is by far the best known moments generalization bound under $L_q$-stability that holds for all $q \geq 2$ and potentially unbounded losses (Celisse & Guedj, 2016; Abou-Moustafa & Szepesvári, 2019):

$$\|R(A(S)) - R_S(A(S))\|_q \lesssim \gamma_q \sqrt{Nq} + \sqrt{\frac{q}{N}}. \tag{6}$$

As one can see that the $L_q$-stability generalization bound in Eq. (6) is significantly inferior to the near-optimal uniform stability generalization bound in Eq. (3) in terms of the overhead on stability coefficient. Such a gap in rate of convergence is indeed unsurprising: the bound in Eq. (6) was derived via more or less directly applying moments or exponential extensions of Efron-Stein inequality to generalization error (Celisse & Guedj, 2016; Abou-Moustafa & Szepesvári, 2019), and thus yields about the same overhead factor on stability coefficient as that of the sub-optimal exponential bound in Eq. (2) for uniformly stable algorithms. In light of these observations, we are naturally motivated to derive sharper exponential generalization bounds for $L_q$-stable algorithms hopefully to match the near-optimal bound in Eq. (3) achievable by uniformly stable algorithms.

## 1.2 OUR CONTRIBUTION

The core contribution of the present work is a set of substantially improved exponential generalization bounds for $L_q$-stable algorithms. The key ingredient of our analysis is a sharper concentration bound on sums of functions of independent random variables under the $L_q$-norm bounded difference conditions, which generalizes a previous counterpart under the uniform bounded difference conditions (Bousquet et al., 2020). With this generic concentration bound in hand, we are able to derive sharper generalization and excess risk bounds for $L_q$-stable learning algorithms that match those best known for uniform stable algorithms. The power of our results is demonstrated through deriving more appealing exponential sparse excess risk bounds for computationally tractable sparsity estimation algorithms (such as IHT). The main results obtained in this work are sketched below:

- In Section 2, we first establish in Theorem 1 an $L_q$-norm inequality for sums of functions of random variables with $L_q$-norm bounded difference. Then equipped with such a general-purpose concentration inequality, we prove in Theorem 2 the following $L_q$-norm generalization bound for $L_q$-stable learning algorithms for all $q \geq 2$:

$$\|R(A(S)) - R_S(A(S))\|_q \lesssim q\gamma_q \log N + M_q \sqrt{\frac{q}{N}},$$

  where $M_q$ is an upper bound of moments $\|\ell(A(S); Z)\|_q$. Compared to Eq. (6), the preceding bound improves the overhead factor on $\gamma_q$ from $\sqrt{N}$ to $\log(N)$. As another consequence of our $L_q$-norm concentration inequality, we further derive in Theorem 3 the following excess risk bound for $L_q$-stable algorithms under $B$-Bernstein-condition with $M$-bounded losses (where $C = M + B$), or $\mu$-quadratic-growth condition with $G$-Lipschitz losses (where $C = \frac{G^2}{\mu}$):

$$\|R(A(S)) - R^* - \Delta_{\text{opt}}\|_q \lesssim \mathbb{E}[\Delta_{\text{opt}}] + q\gamma_q \log(N) + \frac{Cq}{N}.$$

Based on the equivalence between moments and tails, this above result implies an identical deviation optimal risk bound in Eq. (5) for uniformly stable algorithms.

- In Section 3, based on our $L_q$-stability generalization theory, we show in Theorem 4 a novel exponential sparse excess risk bound for inexact $L_0$-estimators. A key insight here is that $L_0$-estimators are in many cases "almost always" stable over any fixed supporting set, and thus can be shown to have $L_q$-stability over the same supporting set, which consequently makes our analysis techniques developed for $L_q$-stable algorithms applicable there. This novel application answers a call by Celisse & Guedj (2016) for extending the range of applicability of the $L_q$-stability theory beyond the unbounded ridge regression problem, and it complements other existing applications of the $L_q$-stability theory including $k$-nearest neighbor classification and $k$-folds cross-validation (Celisse & Mary-Huard, 2018; Abou-Moustafa & Szepesvári, 2019). Last but not least, our improved $L_q$-stability theory can also be readily applied to the above mentioned prior applications to obtain sharper generalization bounds.

## 2 SHARPER EXPONENTIAL BOUNDS FOR $L_q$-STABLE ALGORITHMS

### 2.1 A MOMENT INEQUALITY FOR SUMS OF RANDOM FUNCTIONS

We start by presenting in the following theorem a moment inequality for sums of random functions of $N$ independent random variables that satisfy the $L_q$-norm bounded difference condition. See Appendix B.1 for its proof.

**Theorem 1.** *Let $S = \{Z_1, Z_2, ..., Z_N\}$ be a set of independent random variables valued in $\mathcal{Z}$. Let $g_1, ..., g_N$ be a set of measurable functions $g_i : \mathcal{Z}^N \mapsto \mathbb{R}$ that satisfy the following conditions for any $i \in [N]$:*

- $\mathbb{E}\left[g_i(S) \mid S \setminus Z_i\right] = 0$, *almost surely;*

- $g_i(S)$ *has the following $L_q$-norm bounded difference property with respect to all variables in $S$ except $Z_i$, i.e., $\forall j \neq i$, for all $q \geq 1$:*

$$\left\| g_i(S) - g_i(S^{(j)}) \right\|_q \leq \beta_q.$$

*Then there exists a universal constant $\kappa < 1.271$ such that for all $q \geq 2$,*

$$\left\| \sum_{i=1}^N g_i(S) - \mathbb{E}[g_i(S) \mid Z_i] \right\|_q \leq 4\kappa q N \lceil \log_2 N \rceil \beta_q.$$

*Additionally, if $\|\mathbb{E}[g_i(S) \mid Z_i]\|_q \leq M_q$, then for all $q \geq 2$*

$$\left\| \sum_{i=1}^N g_i(S) \right\|_q \leq 2\sqrt{2\kappa N q} M_q + 4\kappa q N \lceil \log_2 N \rceil \beta_q.$$

**Remark 1.** *Theorem 1 extends the moment inequality of Bousquet et al. (2020, Theorem 4) from under the distribution-free uniform bounded difference property to under the $L_q$-norm bounded difference property which is distribution dependent. Specially if $g_i(S)$ have uniformly bounded difference property, then Theorem 1 reduces to the result of Bousquet et al. (2020, Theorem 4).*

**Remark 2.** *The $L_q$-norm boundedness condition $\|\mathbb{E}[g_i(S) \mid Z_i]\|_q \leq M_q$ in our theorem allows $g_i$ to be potentially unbounded over domain of interest, which is weaker than the corresponding almost sure boundedness condition on $|\mathbb{E}[g_i(S) \mid Z_i]|$ as imposed by Bousquet et al. (2020, Theorem 4).*

### 2.2 GENERALIZATION BOUNDS FOR $L_q$-STABLE ALGORITHMS

As an important consequence of Theorem 1, we can derive er the following main result on the generalization bound of $L_q$-stable learning algorithms. See Appendix B.2 for a proof of this result.

**Theorem 2.** *Let $A : \mathcal{Z}^N \mapsto \mathcal{W}$ be a learning algorithm that has $L_q$-stability by $\gamma_q > 0$ for $q \geq 1$. Suppose that $\|\ell(A(S); Z)\|_q \leq M_q$ for any $Z \in \mathcal{Z}$. Then for all $q \geq 2$,*

$$\|R(A(S)) - R_S(A(S))\|_q \lesssim q\gamma_q \log N + M_q \sqrt{\frac{q}{N}}.$$

**Remark 3.** *The $L_q$-norm boundedness condition $\|\ell(A(S); Z)\|_q \leq M_q$ allows for learning with unbounded losses over, e.g., data distribution with sub-Gaussian or sub-exponential tail bounds.*

To compare with the best-known moments generalization bound in Eq. (6) under the notion of $L_q$-stability, our bound in Theorem 2 substantially improves the overhead factor on $\gamma_q$ from $\sqrt{N}$ to $\log(N)$. Specially when reduced to regime of uniform stability where $\gamma_q \equiv \gamma_u$ for all $q \geq 1$, our result revisits the moments generalization bound in Eq. (3) which is nearly tight, up to logarithmic factors on sample size, in the sense of a lower bound on sums of random functions from Bousquet et al. (2020). More broadly, for any $\delta \in (0, 1)$, suppose that the following exponential stability bound holds with probability at least $1 - \delta$ with a mixture of sub-Gaussian and sub-exponential tails [1] over $S, S^{(i)}, Z$:

$$\left| \ell(A(S); Z) - \ell(A(S^{(i)}); Z) \right| \leq a \log\left(\frac{e}{\delta}\right) + b\sqrt{\log\left(\frac{e}{\delta}\right)}. \tag{7}$$

Then according to the equivalence of tails and moments, as summarized in Lemma 4 (see Appendix A), we must have that $A$ is $L_q$-stable by $\gamma_q = aq + b\sqrt{q}$. Assume that the loss is bounded in $(0, M]$ almost surely over data. Then the $L_q$-norm generalization bound in Theorem 2 combined with Lemma 4 immediately implies the following generalization bound:

$$|R(A(S)) - R_S(A(S))| \lesssim a \log(N) \log^2\left(\frac{1}{\delta}\right) + b \log(N) \log^{1.5}\left(\frac{1}{\delta}\right) + M\sqrt{\frac{\log(1/\delta)}{N}}.$$

Compared with the uniform stability implied tail bound in Eq. (4), the preceding $L_q$-stability bound is nearly identical up to slightly worse confidence tail terms which are caused by the uncertainty of $L_q$-stability with respect to data distribution. We conjecture that such a slight deterioration in tail bounds might possibly be remedied by using the exponential versions of Efron-Stein inequality (Boucheron et al., 2003; Abou-Moustafa & Szepesvári, 2019) instead of the currently used variant in moments. We leave the improvement over poly-logarithmic terms for future investigation.

## 2.3 EXCESS RISK BOUNDS FOR $L_q$-STABLE ALGORITHMS

In addition to the generalization bounds, we further apply Theorem 1 to study the excess risk bounds of an $L_q$-stable learning algorithm which are of particular interest for understanding its population risk minimization performance. Let us denote $W^* := \text{Argmin}_{w \in \mathcal{W}} R(w)$ as the optimal solution set of the population risk. In order to get sharper risk bounds, we need to impose some structural conditions on risk functions. Particularly, the following defined generalized Bernstein condition (Koltchinskii, 2006) is conventionally used with multiple global risk minimizers allowed.

**Assumption 1** (Generalized Bernstein condition). *For some $B > 0$ and for any $w \in \mathcal{W}$, there exists $w^* \in W^*$ such that the following holds:*

$$\mathbb{E}\left[(\ell(w; Z) - \ell(w^*; Z))^2\right] \leq B(R(w) - R(w^*)).$$

We will also consider the quadratic growth condition which is widely used as an alternative condition for establishing fast rates of convergence in learning theory.

**Assumption 2** (Quadratic growth condition). *For some $\mu > 0$ and for any $w \in \mathcal{W}$, there exists $w^* \in W^*$ such that the following holds:*

$$R(w) \geq R^* + \frac{\mu}{2}\|w - w^*\|^2.$$

**Remark 4.** *Clearly, when the loss is $G$-Lipschitz, the quadratic growth condition with parameter $\mu$ implies the Bernstein condition with parameter $B = \frac{2G^2}{\mu}$.*

The following theorem is our main result on the excess risk bound of $L_q$-stable algorithms, which extends the near-optimal exponential risk bounds of Klochkov & Zhivotovskiy (2021) from uniform stable algorithms to $L_q$-stable algorithms. A proof of this result can be found in Appendix B.3.

---

[1] In an exponential tail bound, the terms associated with $\sqrt{\log\left(\frac{1}{\delta}\right)}$ and $\log\left(\frac{1}{\delta}\right)$ are respectively referred to as sub-Gaussian and sub-exponential tails.

**Theorem 3.** *Let $A : \mathcal{Z}^N \mapsto \mathcal{W}$ be a learning algorithm that has $L_q$-stability with parameter $\gamma_q$ for $q \geq 1$.*

*(a) If Assumption 1 holds and $\ell(\cdot; \cdot) \leq M$, then $\forall q \geq 2$,*

$$\| R(A(S)) - R^* - \Delta_{opt} \|_q \lesssim \mathbb{E}[\Delta_{opt}] + q\gamma_q \log(N) + \frac{(M+B)q}{N}.$$

*(b) If Assumption 2 holds and $\ell(\cdot; \cdot)$ is G-Lipschitz with respect to its first argument, then $\forall q \geq 2$,*

$$\| R(A(S)) - R^* - \Delta_{opt} \|_q \lesssim \mathbb{E}[\Delta_{opt}] + q\gamma_q \log(N) + \frac{G^2 q}{\mu N}.$$

**Remark 5.** *Suppose that $A$ satisfies the exponential stability bound in Eq. (7), and thus $A$ has $L_q$-stability by $\gamma_q = aq + b\sqrt{q}$. Then combined with Lemma 4, the $L_q$-norm risk bounds in Theorem 3 suggest that the following exponential tail bound holds:*

$$R(A(S)) - R^* \lesssim \Delta_{opt} + \mathbb{E}[\Delta_{opt}] + a\log(N)\log^2\left(\frac{1}{\delta}\right) + b\log(N)\log^{1.5}\left(\frac{1}{\delta}\right) + \frac{\log(1/\delta)}{N}.$$

**Remark 6.** *In part (a), the $M$-bounded-loss condition is not essential and it can be relaxed to a sub-exponential (or sub-Gaussian) variant by alternatively using the general Bernstein-type inequalities for sums of independent sub-exponential random variables (Vershynin, 2018). Concerning part (b), under the quadratic growth condition, the loss is allowed to be unbounded if it is Lipschitz continuous.*

## 3    APPLICATION TO INEXACT $L_0$-ERM

In this section, we demonstrate an application of our $L_q$-stability and generalization theory to the following problem of high-dimensional stochastic risk minimization under hard sparsity constraint:

$$\min_{w \in \mathcal{W}} R(w) := \mathbb{E}_Z[\ell(w; Z)] \quad \text{subject to } \|w\|_0 \leq k,$$

where $\mathcal{W} \subseteq \mathbb{R}^d$ the cardinality constraint $\|w\|_0 \leq k$ for $k \ll d$ is imposed for enhancing the interpretability and learnability of model in situations where there are no clear favourite explanatory variables, or the model is over-parameterized. We consider the following $L_0$-ERM problem over training set $S = \{Z_i\}_{i \in [N]}$:

$$w^*_{S,k} = \operatorname*{arg\,min}_{\|w\|_0 \leq k} \left\{ R_S(w) := \frac{1}{N} \sum_{i=1}^{N} \ell(w; Z_i) \right\}. \tag{8}$$

Since the problem is known to be NP-hard (Natarajan, 1995) in general, it is computationally intractable to solve it exactly in general cases. Alternatively, we consider the inexact $L_0$-ERM oracle as a meta-algorithm outlined in Algorithm 1. In order to avoid assuming unrealistic conditions like restricted isometry property (RIP), it is typically needed to allow sparsity level relaxation for approximate algorithms like IHT to achieve favorable converge behavior (Jain et al., 2014; Shen & Li, 2017; Yuan et al., 2018; Murata & Suzuki, 2018). Therefore, we are particularly interested in

---

**Algorithm 1:** Inexact $L_0$-ERM Oracle

---

**Input** : A training data set $S = \{Z_i\}_{i \in [N]}$ and the desired sparsity level $k$.
**Output:** $\tilde{w}_{S,k}$.
Compute an inexact $k$-sparse $L_0$-ERM estimation $\tilde{w}_{S,k}$ such that

- $\tilde{w}_{S,k}$ is optimal over its support $\tilde{J} = \text{supp}(\tilde{w}_{S,k})$, i.e., $\tilde{w}_{S,k} = \arg\min_{w \in \mathcal{W}, \text{supp}(w) \subseteq \tilde{J}} R_S(w)$;

- $\tilde{w}_{S,k}$ attains certain $\bar{k}$-sparse sub-optimality level $\Delta_{\bar{k}, \text{opt}} \geq 0$ for some $\bar{k} \leq k$ such that
  $R_S(\tilde{w}_{S,k}) - R_S(w^*_{S,\bar{k}}) \leq \Delta_{\bar{k}, \text{opt}}$.

---

the inexact $L_0$-ERM oracle with $\bar{k}$-sparse sub-optimality $\Delta_{\bar{k},\mathrm{opt}} \geq 0$ for some $\bar{k} \leq k$ such that the output $\tilde{w}_{S,k}$ of Algorithm 1 satisfies

$$R_S(\tilde{w}_{S,k}) - R_S(w^*_{S,\bar{k}}) \leq \Delta_{\bar{k},\mathrm{opt}}.$$

It is typical that $\Delta_{\bar{k},\mathrm{opt}}$ is a random value over the training set $S$. For example, the sub-optimality guarantees of IHT for empirical risk usually hold with high probability over training data (Jain et al., 2014). Let $w^*_{\bar{k}} := \arg\min_{\|w\|_0 \leq \bar{k}} R(w)$ be the $\bar{k}$-sparse minimizer of population risk for some $\bar{k} \leq k$. We are interested in deriving exponential upper bounds for the $\bar{k}$-sparse excess risk given by $R(\tilde{w}_{S,k}) - R(w^*_{\bar{k}})$.

Our analysis also relies on the conditions of Restricted Strong Convexity (RSC) which extends the concept of strong convexity to the analysis of sparsity recovery methods (Bahmani et al., 2013; Blumensath & Davies, 2009; Jain et al., 2014; Yuan et al., 2020).

**Definition 3** (Restricted Strong Convexity). *For any sparsity level $1 \leq s \leq d$, we say a function $f$ is restricted $\mu_s$-strongly convex if there exists some $\mu_s > 0$ such that*

$$f(w) - f(w') - \langle \nabla f(w'), w - w' \rangle \geq \frac{\mu_s}{2} \|w - w'\|^2, \ \ \forall \|w - w'\|_0 \leq s.$$

*Specially when $s = d$, we say $f$ is $\mu$-strongly convex if it is $\mu_d$-strongly convex.*

The following basic assumptions will be used in our theoretical analysis.

**Assumption 3.** *The loss function $\ell(\cdot; \cdot)$ is convex and $G$-Lipschitz with respect to its first argument.*

**Assumption 4.** *The population risk $R$ is $\mu$-strongly convex and the empirical risk $R_S$ is $\mu_k$-strongly convex with probability at least $1 - \delta_N$ over sample $S$ for some $\delta_N \in (0,1)$.*

**Assumption 5.** *The domain of interest is uniformly bounded such that $\|w\| \leq D, \forall w \in \mathcal{W}$.*

**Remark 7.** *Assumption 3 is common in the study of algorithmic stability and generalization theory. Assumption 4 is conventional in the sparsity recovery analysis of $L_0$-ERM. Assumption 5 is needed for establishing the $L_q$-stability of $L_0$-ERM in Lemma 1 to follow. Similar conditions have also been assumed in the prior work of Yuan & Li (2022).*

Let $w^* := \arg\min_{w \in \mathcal{W}} R(w)$ be the global minimizer which is unique due to the strong convexity of $R$. For a given index set $J \subseteq [d]$, let us consider the following restrictive estimator over $J$:

$$w^*_{S|J} := \arg\min_{w \in \mathcal{W}, \mathrm{supp}(w) \subseteq J} R_S(w). \tag{9}$$

We first present the following lemma that guarantees the $L_q$-stability of $w^*_{S|J}$ for any fixed $J$ with $|J| = k$. See Appendix C.1 for its proof.

**Lemma 1.** *Assume that Assumptions 3, 4 and 5 hold and $\frac{\log(1/\delta_N)}{\log(N)} \geq 2$. Let $J \subseteq [d], |J| = k$ be a set of indices of cardinality $k$. Then for any $q \geq 2$, the oracle estimator $w^*_{S|J}$ has $L_q$-stability with parameter*

$$\gamma_q = \frac{1}{N}\left(\frac{4G^2}{\mu_k} + 2GD\right) + \frac{2GD\log(N)q}{\log(1/\delta_N)}.$$

**Remark 8.** *For sparse linear regression models, it can be verified based on the result by Agarwal et al. (2012, Lemma 6) that Assumptions 4 holds with $\delta_N = e^{-c_0 N}$ for some universal positive constant $c_0$. Then we have $\frac{\log(1/\delta_N)}{\log(N)} = \frac{c_0 N}{\log(N)} \geq 2$ for sufficiently large $N$, and Lemma 1 implies that $\gamma_q \lesssim \frac{1}{N}\left(\frac{G^2}{\mu_k} + \frac{GD\log(N)q}{c_0}\right)$ for all $q \geq 2$.*

The following theorem is our main result on the sparse excess risk of the inexact $L_0$-ERM oracle as defined in Algorithm 1. See Appendix C.2 for its proof which is stimulated by that of Theorem 3.

**Theorem 4.** *Suppose that Assumptions 3, 4, 5 hold. Assume that $\frac{\log(1/\delta_N)}{\log(N)} \geq 2$. Then for any $\delta \in (0, e^{-1})$, the following $\bar{k}$-sparse excess risk bound holds with probability at least $1 - \delta$ over the random draw of $S$:*

$$R(\tilde{w}_{S,k}) - R(w^*_{\bar{k}})$$

$$\lesssim \frac{GD\left(k\log\left(\frac{ed}{k}\right) + \log\left(\frac{e}{\delta}\right)\right)^2 \log^2(N)}{\log(1/\delta_N)} + \left(\log(N)\left(\frac{G^2}{\mu_k} + GD\right) + \frac{G^2}{\mu}\right)\frac{k\log\left(\frac{ed}{k}\right) + \log\left(\frac{e}{\delta}\right)}{N}$$

$$+ G\sqrt{\frac{\left(k\log\left(\frac{ed}{k}\right) + \log\left(\frac{e}{\delta}\right)\right)\left(R(w^*_{\bar{k}}) - R(w^*)\right)}{N\mu}} + \Delta_{\bar{k},opt} + \mathbb{E}\left[\Delta_{\bar{k},opt}\right].$$

**Remark 9.** *For the IHT-style algorithms, the sparse optimization sub-optimality $\Delta_{\bar{k},opt}$ can be arbitrarily small (with high probability) after sufficient rounds of iteration (Jain et al., 2014).*

Specially for sparse linear regression models in which Assumptions 4 holds with $\delta_N = e^{-c_0 N}$ (Agarwal et al., 2012), we have that $\frac{\log(1/\delta_N)}{\log(N)} = \frac{c_0 N}{\log(N)} \geq 2$ can always be fulfilled for sufficiently large sample size $N$, and the sparse excess risk bound in Theorem 4 roughly scales as

$$R(\tilde{w}_{S,k}) - R(w_{\bar{k}}^*) \lesssim \frac{(k\log(d) + \log(1/\delta))^2 \log^2(N)}{N}$$
$$+ \sqrt{\frac{(k\log(d) + \log(1/\delta))(R(w_{\bar{k}}^*) - R(w^*))}{N}} + \Delta_{\bar{k},\text{opt}} + \mathbb{E}\left[\Delta_{\bar{k},\text{opt}}\right].$$

Generally for misspecified sparsity models, the dominant rate in the above bound matches the $\mathcal{O}\left(\frac{1}{\sqrt{N}}\right)$ sparse excess risk bound of Yuan & Li (2022, Theorem 1) for IHT under similar conditions. Compared to the $\mathcal{O}\left(\frac{1}{N}\right)$ bound available in that paper (Yuan & Li, 2022, Theorem 3), the preceding bound is generally slower in rate but more broadly applicable without imposing any strong-signal or bounded-loss conditions as required in the analysis of Yuan & Li (2022).

For well-specified $\bar{k}$-sparse models such that $R(w_{\bar{k}}^*) = R(w^*)$, i.e., the population minimizer is truly $\bar{k}$-sparse, the preceding bound improves to

$$R(\tilde{w}_{S,k}) - R(w_{\bar{k}}^*) \lesssim \frac{(k\log(d) + \log(1/\delta))^2 \log^2(N)}{N} + \Delta_{\bar{k},\text{opt}} + \mathbb{E}\left[\Delta_{\bar{k},\text{opt}}\right].$$

Therefore, our bound is more appealing in the sense that it naturally adapts to well-specified models to attain an improved $\mathcal{O}\left(\frac{1}{N}\right)$ rate. In contrast, the regularization technique used by Yuan & Li (2022, Theorem 1) needs an optimal choice of penalty strength of scale $O(\frac{1}{\sqrt{N}})$ which leads to an overall slow rate of convergence, though the analysis is relatively simpler.

We further comment on the role of $L_q$-stability played in deriving the improved bound of Theorem 4. The $O(\frac{1}{N})$ fast-rate component of the bound is indeed rooted from the $L_q$-stability coefficient as established in Lemma 1 and an application of Lemma 6. The relatively slow $O(\frac{1}{\sqrt{N}})$ component, which is controlled by the oracle factor $R(w_{\bar{k}}^*) - R(w^*)$, is mainly due to a careful analysis customized for handling the combinatorial optimization nature of $L_0$-ERM. Such a slow-rate term would be vanished if the global minimizer $w^*$ is truly $\bar{k}$-sparse. Therefore, we confirm that the fast-rate component attributes to our $L_q$-stability theory, while the slow but adaptive rate component mainly attributes to the optimization property of $L_0$-ERM. Finally, we comment in passing that our improved $L_q$-stability theory can also be applied to some prior applications such as unbounded ridge regression and $k$-folds cross-validation to obtain sharper generalization bounds.

## 4 CONCLUSION

In this paper, we presented an improved generalization theory for $L_q$-stable learning algorithms. There exits a clear discrepancy between the recently developed near-optimal generalization bounds for uniformly stable algorithms and the best known yet sub-optimal bounds for $L_q$-stable algorithms. Aiming at closing such a theoretical gap, we for the first time derived a set of near-optimal exponential generalization bounds for $L_q$-stable algorithms that match those of uniformly stable algorithms. As a concrete application of our $L_q$-stability theory, we have applied the developed analysis tools to derive strong exponential risk bounds for inexact sparsity-constrained ERM estimators under milder conditions. To conclude, $L_q$-stable algorithms generalize almost as fast as uniformly stable algorithms, though the distribution-dependent notion of $L_q$-stability is weaker than uniform stability.

## ACKNOWLEDGMENTS AND DISCLOSURE OF FUNDING

The authors would like to thank the anonymous Reviewers and Area Chairs for their insightful comments which are truly helpful for improving this paper. Xiao-Tong Yuan is funded in part by the National Key Research and Development Program of China under Grant No.2018AAA0100400, and in part by the Natural Science Foundation of China (NSFC) under Grant No.U21B2049, No.61936005 and No.61876090.

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

## A PRELIMINARIES

In this section, we collect some preliminary results that will be used in our analysis. We start by introducing the following $L_q$-norm generalization of the celebrated Efron-Stein inequality, which is a corollary of Boucheron et al. (2005, Theorem 2).

**Proposition 1** (Generalized Efron-Stein inequality (Celisse & Guedj, 2016)). *Let $S = \{Z_1, ..., Z_N\}$ be a set of independent random variables valued in $\mathcal{Z}$ and $g : \mathcal{Z}^N \mapsto \mathbb{R}$ be some measurable function. Then there exists a universal constant $\kappa < 1.271$ such that for all $q \geq 2$,*

$$\|g(S) - \mathbb{E}[g(S)]\|_q \leq \sqrt{2\kappa q}\sqrt{\left\|\sum_{i=1}^{N} \left(g(S) - g(S^{(i)})\right)^2\right\|_{q/2}}.$$

The following result is an immediate consequence of Proposition 1 when applied to sum of independent random variables, which revisits a version of Marcinkiewicz-Zygmund inequality (Chow & Teicher, 2003).

**Proposition 2.** *Let $Z_1, ..., Z_N$ be a set of independent centered random variables. Then for all $q \geq 2$,*

$$\left\|\sum_{i=1}^{N} Z_i\right\|_q \leq 2\sqrt{2\kappa q}\sqrt{\left\|\sum_{i=1}^{N} Z_i^2\right\|_{q/2}}.$$

The following lemma is simple yet useful in our analysis.

**Lemma 2.** *Let $S = \{Z_1, ..., Z_N\}$ be a set of independent random variables valued in some measure space $\mathcal{Z}$ and $g : \mathcal{Z}^N \mapsto \mathbb{R}$ be some measurable function. Then for all $I \subseteq [N]$ and $q \geq 1$, we have*

$$\|g(S_I)\|_q \leq \|g(S)\|_q = \|\|g\|_q(S_I)\|_q.$$

*Proof.* Recall $g(S_I) = \mathbb{E}[g(S) \mid S_I]$. Then using Jensen's inequality we can show that

$$\|g(S_I)\|_q = (\mathbb{E}\left[|\mathbb{E}[g(S) \mid S_I]|^q\right])^{1/q} \leq (\mathbb{E}\left[\mathbb{E}[|g(S)|^q \mid S_I]\right])^{1/q} = (\mathbb{E}[|g(S)|^q])^{1/q} = \|g(S)\|_q.$$

By definition we can also express $\|g(S)\|_q = (\mathbb{E}\left[\mathbb{E}[|g(S)|^q \mid S_I]\right])^{1/q} = \|\|g(S)\|_q(S_I)\|_q$. $\qquad\square$

As a direct consequence of Lemma 2, the following result indicates that conditional expectation does not expand the differences in $L_q$-norm.

**Lemma 3.** *Let $S = \{Z_1, ..., Z_N\}$ be a set of independent random variables valued in some measure space $\mathcal{Z}$ and $g : \mathcal{Z}^N \mapsto \mathbb{R}$ be some measurable function. Let $I \subseteq [N]$ be an index set. Then for all $i \in I$ and $q \geq 1$,*

$$\left\|g(S_I) - g(S_I^{(i)})\right\|_q \leq \left\|g(S) - g(S^{(i)})\right\|_q.$$

*Proof.* For each $i \in I$, by applying Lemma 2 to $g(S) - g(S^{(i)})$ we can show that $\|g(S_I) - g(S_I^{(i)})\|_q \leq \|g(S) - g(S^{(i)})\|_q$, which gives the desired result. $\qquad\square$

We also need the following lemma about the equivalence between tails and moments (see, e.g., Bousquet et al., 2020).

**Lemma 4.** *Let $Y$ be a real-valued random variable.*

- *Suppose that $Y$ satisfies the following inequality for some $a, b \geq 0$ with probability at least $1 - \delta$ for any $\delta \in (0, 1)$,*

$$|Y| \leq a \log\left(\frac{e}{\delta}\right) + b\sqrt{\log\left(\frac{e}{\delta}\right)}.$$

*Then, for any $q \geq 1$ it holds that*

$$\|Y\|_q \leq 3aq + 9b\sqrt{q}.$$

- *Suppose that $Y$ satisfies $\|Y\|_q \leq f(q)$ for any $1 \leq q_l \leq q < q_u$ and some non-negative real function $f$. Then the following holds with probability at least $1 - \delta$ for any $\delta \in (e^{1-q_u}, e^{1-q_l}]$:*

$$|Y| \leq ef\left(\log\left(\frac{e}{\delta}\right)\right).$$

*Proof.* We only prove the second part which slightly generalizes the corresponding result of Bousquet et al. (2020, Lemma 1). For any $\delta \in (e^{1-q_u}, e^{1-q_l}]$, we choose $q = \log(e/\delta) \in [q_l, q_u)$. Using the condition $\|Y\|_q \leq f(q)$ and Markov's inequality yields

$$\mathbb{P}\left(|Y| > ef\left(\log\left(\frac{e}{\delta}\right)\right)\right) \leq \mathbb{P}\left(|Y| > e\|Y\|_q\right) \leq \frac{\mathbb{E}[|Y|^q]}{e^q\|Y\|_q^q} = \frac{\delta}{e} \leq \delta.$$

This proves the desired bound in the second part. $\qquad\square$

**Remark 10.** *Suppose that for any $\delta \in (0,1)$, the following inequality holds with probability at least $1 - \delta$ over $S, S^{(i)}, Z$:*

$$\left|\ell(A(S); Z) - \ell(A(S^{(i)}); Z)\right| \leq a\log\left(\frac{e}{\delta}\right) + b\sqrt{\log\left(\frac{e}{\delta}\right)}.$$

*Then according to the first part of Lemma 4 we have that $A$ is $L_q$-stable by $\gamma_q = 3aq + 9b\sqrt{q}$.*

**Remark 11.** *Specially if $q_l = 1$ and $q_u = \infty$ is allowed, then the second bound in Lemma 4 holds with an arbitrary tail bound $\delta \in (0,1)$.*

Finally, we present the following technical lemma about self-bounding inequalities to be used for showing fast rates of excess risk bounds under Bernstein or quadratic growth conditions.

**Lemma 5.** *Let $x, a, b, c$ be a set of non-negative quantities satisfying $x \leq a + \sqrt{b(x+c)}$. Then it must hold that $x \leq \frac{3a+2b+c}{2}$.*

*Proof.* If $x \leq a$, then the claim holds trivially. In the complementary case of $x > a$, by condition we must have $(x-a)^2 \leq b(x+c)$, which then implies

$$x \leq \frac{2a+b+2\sqrt{b(a+c)}}{2} \leq \frac{3a+2b+c}{2},$$

where we have used the basic fact $2\sqrt{b(a+c)} \leq b + a + c$. $\qquad\square$

# B   PROOFS FOR SECTION 2

## B.1   PROOF OF THEOREM 1

The proof is a generalization of the sample-splitting arguments of Feldman & Vondrák (2019); Bousquet et al. (2020) under the considered property of $L_q$-norm bounded difference. For the sake of completeness, we reproduce below the relatively simpler arguments of Bousquet et al. (2020, Theorem 4), with proper modifications made to adapt to our setting via using the generalized Efron-Stein inequality in places of McDiarmid's inequality.

*Proof of Theorem 1.* Consider $k$ such that $2^{k-1} < N \leq 2^k$. If $N < 2^k$, we pad the training set $S$ with extra zero-functions so that $N = 2^k$. Consider the partition $\mathcal{I}_0, \mathcal{I}_1, ..., \mathcal{I}_k$ of $[N]$ given by

$$\mathcal{I}_0 = \{\{1\}, ..., \{2^k\}\}, \quad \mathcal{I}_1 = \{\{1,2\}, \{3,4\}..., \{2^k-1, 2^k\}\}, \quad \mathcal{I}_k = \{\{1, ..., 2^k\}\}.$$

For any $i \in [N]$ and $l = 0, ..., k$, we denote by $I^l(i) \in \mathcal{I}_l$ the only set from $\mathcal{I}_l$ that contains $i$ and consider the following random variables

$$g_i^l = \mathbb{E}\left[g_i \mid Z_i, S_{\overline{I^l(i)}}\right].$$

In particular, $g_i^0 = g_i$ and $g_i^k = \mathbb{E}[g_i \mid Z_i]$. Clearly we have the following telescope sum:

$$g_i = \sum_{l=0}^{k-1}(g_i^l - g_i^{l+1}) + \mathbb{E}[g_i \mid Z_i].$$

It follows that

$$\left\| \sum_{i=1}^{N} g_i - \mathbb{E}[g_i \mid Z_i] \right\|_q \leq \sum_{l=0}^{k-1} \left\| \sum_{i=1}^{N} g_i^l - g_i^{l+1} \right\|_q. \tag{10}$$

We need to upper bound the right hand side of the above inequality. To this end, it can be verified that

$$g_i^{l+1} = \mathbb{E}\left[ g_i \mid Z_i, S_{\overline{I^{l+1}(i)}} \} \right] = \mathbb{E}\left[ g_i^l \mid Z_i, S_{\overline{I^{l+1}(i)}} \right].$$

Since $g_i$ has a bounded $L_q$-difference by $\beta_q$ with respect to all variables except the $i$-th variable, it is known from Lemma 3 that so is $g_i^l$ for each $l = 0, ..., k$. Conditioned on $Z_i, S_{\overline{I^{l+1}(i)}}$, invoking Proposition 1 to $g_i^l$ yields

$$\|g_i^l - g_i^{l+1}\|_q \left( Z_i, S_{\overline{I^{l+1}(i)}} \right) \leq \sqrt{2\kappa q 2^l} \beta_q,$$

as there are $2^l$ indices in $I^{l+1}(i) \setminus I^l(i)$. It follows from Lemma 2 that

$$\|g_i^l - g_i^{l+1}\|_q = \left\| \|g_i^l - g_i^{l+1}\|_q(Z_i, S_{\overline{I^{l+1}(i)}}) \right\|_q \leq \sqrt{2\kappa q 2^l} \beta_q.$$

Now consider any $I^l \in \mathcal{I}_l$. Since for each $i \in I_l$, $g_i^l - g_i^{l+1}$ depends only on $Z_i, S_{\overline{I^l}}$, these terms are independent and centered conditioned on $S_{\overline{I^l}}$. Therefore, applying Proposition 2 yields

$$\left\| \sum_{i \in I^l} g_i^l - g_i^{l+1} \right\|_q \left( S_{\overline{I^l}} \right) \leq 2\sqrt{2\kappa q 2^l} \times \sqrt{2\kappa q 2^l} \beta_q = 4\kappa q 2^l \beta_q,$$

which according to Lemma 2 implies that

$$\left\| \sum_{i \in I^l} g_i^l - g_i^{l+1} \right\|_q \leq 4\kappa q 2^l \beta_q.$$

Then based on the triangle inequality we get

$$\left\| \sum_{i \in [N]} g_i^l - g_i^{l+1} \right\|_q \leq \sum_{I^l \in \mathcal{I}_l} \left\| \sum_{i \in I^l} g_i^l - g_i^{l+1} \right\|_q \leq 2^{k-l} \times 4\kappa q 2^l \beta_q = 4\kappa q 2^k \beta_q < 4\kappa q N \beta_q.$$

Finally, the right hand side of Eq. (10) can be bounded as

$$\left\| \sum_{i=1}^{N} g_i - \mathbb{E}[g_i \mid Z_i] \right\|_q \leq \sum_{l=0}^{k-1} \left\| \sum_{i=1}^{N} g_i^l - g_i^{l+1} \right\|_q \leq 4\kappa q N \lceil \log_2 N \rceil \beta_q, \tag{11}$$

which gives the first desired bound. In view of Eq. (11) and the triangle inequality we have

$$\left\| \sum_{i=1}^{N} g_i \right\|_q \leq \left\| \sum_{i=1}^{N} \mathbb{E}[g_i \mid Z_i] \right\|_q + 4\kappa q N \lceil \log_2 N \rceil \beta_q. \tag{12}$$

Since $\|\mathbb{E}[g_i(S) \mid Z_i]\|_q \leq M_q$ and $\mathbb{E}[g_i(S) \mid S \setminus Z_i] = 0$, it follows from Proposition 2 that the first term at the right hand side of Eq. (12) can be bounded as

$$\left\| \sum_{i=1}^{N} \mathbb{E}[g_i \mid Z_i] \right\|_q \leq 2\sqrt{2\kappa N q} M_q. \tag{13}$$

The second desired bound is obtained by plugging Eq. (13) into Eq. (12). $\qquad \square$

### B.2    PROOF OF THEOREM 2

The proof technique follows that of Bousquet et al. (2020, Lemma 7) developed for uniformly stable algorithms, with natural adaptation to the distribution-dependent notion of $L_q$-stability.

*Proof.* Let us consider

$$h_i(S) := R(A(S)) - \ell(A(S); Z_i), \quad g_i(S) = \mathbb{E}_{Z_i'}\left[ R(A(S^{(i)})) - \ell(A(S^{(i)}); Z_i) \right].$$

Then the $L_q$-norm of the generalization gap can be bounded as

$$\|R(A(S)) - R_S(A(S))\|_q = \frac{1}{N} \left\| \sum_{i=1}^N h_i(S) \right\|_q \leq \frac{1}{N} \left( \underbrace{\left\| \sum_{i=1}^N g_i(S) \right\|_q}_{A} + \underbrace{\left\| \sum_{i=1}^N (h_i(S) - g_i(S)) \right\|_q}_{B} \right). \tag{14}$$

We next respectively upper bound the two terms $A$ and $B$ in Eq. (14). To bound the term $A$, by definition it holds that $\mathbb{E}[g_i(S) \mid S \setminus Z_i] = 0$. Based on the triangle inequality we can show that

$$\begin{aligned}
\|\mathbb{E}[g_i(S) \mid Z_i]\|_q &\leq \|g_i(S)\|_q \\
&= \left\| \mathbb{E}_{Z_i'}[\mathbb{E}_Z[\ell(A(S^{(i)}); Z)]] - \mathbb{E}_{Z_i'}\left[ \ell(A(S^{(i)}); Z_i) \right] \right\|_q \\
&\leq \|\ell(A(S^{(i)}); Z)\|_q + \|\ell(A(S^{(i)}); Z_i)\|_q \leq 2M_q,
\end{aligned}$$

where in the first and second inequalities we have twice used Lemma 2. Next we further show that $g_i$ has a bounded $L_q$-norm difference by $2\gamma_q$ with respect to all variables in $S$ except $Z_i$. Indeed, for each $j \neq i$ it can be verified that

$$\begin{aligned}
&\left\| g_i(S) - g_i(S^{(j)}) \right\|_q \\
&\leq \left\| \mathbb{E}_{Z_i'}\left[ R(A(S^{(i)})) - R(A((S^{(i)})^{(j)})) \right] \right\|_q + \left\| \mathbb{E}_{Z_i'}\left[ \ell(A(S^{(i)}); Z_i) - \ell(A((S^{(i)})^{(j)}); Z_i) \right] \right\|_q \\
&= \left\| \mathbb{E}_{Z_i'}\mathbb{E}_Z[\ell(A(S^{(i)}); Z) - \ell(A((S^{(i)})^{(j)}); Z)] \right\|_q + \left\| \mathbb{E}_{Z_i'}[\ell(A(S^{(i)}); Z_i) - \ell(A((S^{(i)})^{(j)}); Z_i)] \right\|_q \\
&\leq \left\| \ell(A(S^{(i)}); Z) - \ell(A((S^{(i)})^{(j)}); Z) \right\|_q + \left\| \ell(A(S^{(i)}); Z_i) - \ell(A((S^{(i)})^{(j)}); Z_i) \right\|_q \leq 2\gamma_q,
\end{aligned}$$

where in the last but one inequality we have used Lemma 2, while in the last equality we have used the $L_q$-stability assumption on the algorithm $A$. Therefore, $\{g_i\}$ satisfy the conditions of Theorem 1 and thus

$$A = \left\| \sum_{i=1}^N g_i(S) \right\|_q \leq 4\sqrt{2\kappa N q} M_q + 8\kappa q N \lceil \log_2 N \rceil \gamma_q. \tag{15}$$

Now we proceed to bound the term $B$. It can be verified that

$$\begin{aligned}
B &\leq \left\| \sum_{i=1}^N \mathbb{E}_{Z_i'}\left[ R(A(S)) - R(A(S^{(i)})) \right] \right\|_q + \left\| \sum_{i=1}^N \mathbb{E}_{Z_i'}\left[ \ell(A(S); Z_i) - \ell(A(S^{(i)}); Z_i) \right] \right\|_q \\
&= \left\| \sum_{i=1}^N \mathbb{E}_{Z_i'}\mathbb{E}_Z\left[ \ell(A(S); Z) - \ell(A(S^{(i)}); Z) \right] \right\|_q + \left\| \sum_{i=1}^N \mathbb{E}_{Z_i'}\left[ \ell(A(S); Z_i) - \ell(A(S^{(i)}); Z_i) \right] \right\|_q \\
&\leq \sum_{i=1}^N \left\| \ell(A(S); Z) - \ell(A(S^{(i)}); Z) \right\|_q + \sum_{i=1}^N \left\| \ell(A(S); Z_i) - \ell(A(S^{(i)}); Z_i) \right\|_q \leq 2N\gamma_q,
\end{aligned} \tag{16}$$

where in the last but one inequality we have used Lemma 2, and in the last equality we have used the $L_q$-stability assumption. Plugging bounds Eq. (15) and Eq. (16) into Eq. (14) and preserving leading terms yields the desired result. □

### B.3 PROOF OF THEOREM 3

We need the following lemma which plays a fundamental role in proving the main result.

**Lemma 6.** *Let $A : \mathcal{Z}^N \mapsto \mathcal{W}$ be a learning algorithm that has $L_q$-stability by $\gamma_q$ for $q \geq 1$. Suppose that $\|\ell(A(S); Z)\|_q \leq M_q$ for any $Z \in \mathcal{Z}$. Let $S'$ be an independent copy of $S$. Then the following bound holds for all $q \geq 2$:*

$$\left\| R(A(S)) - R_S(A(S)) - \mathbb{E}[R(A(S))] + \frac{1}{N} \sum_{i=1}^{N} \mathbb{E}[\ell(A(S'); Z_i) \mid Z_i] \right\|_q \lesssim q\gamma_q \log(N).$$

*Proof.* Let us again consider $g_i(S) = \mathbb{E}_{Z_i'}\left[ R(A(S^{(i)})) - \ell(A(S^{(i)}); Z_i) \right]$. Then using similar proof arguments to those of Theorem 2 we can show that

$$\left\| N(R(A(S)) - R_S(A(S))) - \sum_{i=1}^{N} g_i(S) \right\|_q$$

$$\leq \left\| \sum_{i=1}^{N} \mathbb{E}_{Z_i'}\left[ R(A(S)) - R(A(S^{(i)})) \right] \right\|_q + \left\| \sum_{i=1}^{N} \mathbb{E}_{Z_i'}\left[ \ell(A(S); Z_i) - \ell(A(S^{(i)}); Z_i) \right] \right\|_q$$

$$= \left\| \sum_{i=1}^{N} \mathbb{E}_{Z_i'}\mathbb{E}_Z\left[ \ell(A(S); Z) - \ell(A(S^{(i)}); Z) \right] \right\|_q + \left\| \sum_{i=1}^{N} \mathbb{E}_{Z_i'}\left[ \ell(A(S); Z_i) - \ell(A(S^{(i)}); Z_i) \right] \right\|_q$$

$$\leq \sum_{i=1}^{N} \left\| \ell(A(S); Z) - \ell(A(S^{(i)}); Z) \right\|_q + \sum_{i=1}^{N} \left\| \ell(A(S); Z_i) - \ell(A(S^{(i)}); Z_i) \right\|_q \leq 2N\gamma_q,$$

which implies

$$\left\| R(A(S)) - R_S(A(S)) - \frac{1}{N} \sum_{i=1}^{N} g_i(S) \right\|_q \leq 2\gamma_q.$$

Also, $g_i(S)$ satisfies the conditions of Theorem 1 with $\beta_q = 2\gamma_q$ and it follows from the second bound of Theorem 1 that for all $q \geq 2$,

$$\left\| \frac{1}{N} \sum_{i=1}^{N} (g_i(S) - \mathbb{E}[g_i(S) \mid Z_i]) \right\|_q \leq 8\kappa q\gamma_q \lceil \log_2 N \rceil.$$

Combining the above two yields

$$\left\| R(A(S)) - R_S(A(S)) - \frac{1}{N} \sum_{i=1}^{N} \mathbb{E}[g_i(S) \mid Z_i] \right\|_q \lesssim q\gamma_q \log(N).$$

The desired result follows by noting that

$$\mathbb{E}[g_i(S) \mid Z_i] = \mathbb{E}[R(A(S'))] - \mathbb{E}[\ell(A(S'); Z_i) \mid Z_i] = \mathbb{E}[R(A(S))] - \mathbb{E}[\ell(A(S'); Z_i) \mid Z_i].$$

This completes the proof. $\square$

With Lemma 6 in place, we are ready to prove the main result of Theorem 3.

*Proof of Theorem 3.* Consider any $w^* \in W^*$. It is standard to decompose and bound the excess risk as

$$
\begin{aligned}
&R(A(S)) - R^* \\
=&R(A(S)) - R_S(A(S)) + R_S(A(S)) - R_S(w^*) + R_S(w^*) - R^* \\
\leq&\Delta_{\mathrm{opt}} + R(A(S)) - R_S(A(S)) - (R^* - R_S(w^*)) \\
=&\Delta_{\mathrm{opt}} + \Gamma(S) + \mathbb{E}[R(A(S))] - \frac{1}{N} \sum_{i=1}^{N} \mathbb{E}[\ell(A(S'); Z_i) \mid Z_i] - (R^* - R_S(w^*)),
\end{aligned}
\tag{17}
$$

where

$$\Gamma(S) = R(A(S)) - R_S(A(S)) - \mathbb{E}[R(A(S))] + \frac{1}{N}\sum_{i=1}^{N}\mathbb{E}[\ell(A(S'); Z_i) \mid Z_i].$$

Since we have the freedom to choose $w^*$, let us specify it in the above as $w^*(S') \in W^*$ which is the minimizer that satisfies the Bernstein condition in Assumption 1 associated with $A(S')$. Then, it follows from Eq. (17) that

$$R(A(S)) - R^* - \Delta_{\mathrm{opt}}$$

$$\leq \Gamma(S) + \mathbb{E}[R(A(S))] - \frac{1}{N}\sum_{i=1}^{N}\mathbb{E}[\ell(A(S'); Z_i) \mid Z_i] - (R^* - \mathbb{E}[R_S(w^*(S')) \mid S]).$$

Consequently,

$$\|R(A(S)) - R^* - \Delta_{\mathrm{opt}}\|_q$$

$$\leq \|\Gamma(S)\|_q + \left\| \frac{1}{N}\sum_{i=1}^{N}\mathbb{E}\left[\ell(w^*(S'); Z_i) - \ell(A(S'); Z_i) \mid Z_i\right] - (R^* - \mathbb{E}[R(A(S'))])\right\|_q$$

$$\overset{\zeta_1}{\lesssim} q\gamma_q \log(N) + \underbrace{\left\| \frac{1}{N}\sum_{i=1}^{N}\mathbb{E}\left[\ell(w^*(S'); Z_i) - \ell(A(S'); Z_i) \mid Z_i\right] - (R^* - \mathbb{E}[R(A(S'))])\right\|_q}_{T}, \qquad (18)$$

where in "$\zeta_1$" we have applied Lemma 6 to obtain $\|\Gamma(S)\|_q \leq q\gamma_q \log(N)$, and the fact $\mathbb{E}[R(A(S))] = \mathbb{E}[R(A(S'))]$.

*Part (a)*: To bound the term $T$, using Bernstein's inequality for sum of independent bounded variables [2] together with the generalized Bernstein condition we can show (see the proof arguments of Klochkov & Zhivotovskiy (2021, Theorem 1.1) for the details) that

$$T \lesssim \sqrt{\frac{qB\mathbb{E}[R(A(S)) - R^*]}{N}} + \frac{qM}{N} = \sqrt{\frac{qB(\mathbb{E}[R(A(S)) - R^* - \Delta_{\mathrm{opt}}] + \mathbb{E}[\Delta_{\mathrm{opt}}])}{N}} + \frac{qM}{N}$$

$$\leq \sqrt{\frac{qB(\|R(A(S)) - R^* - \Delta_{\mathrm{opt}}\|_q + \mathbb{E}[\Delta_{\mathrm{opt}}])}{N}} + \frac{qM}{N},$$

where the last inequality is due to Jensen's inequality. Therefore, combining the above and Eq. (18) yields that for some universal constant $C$:

$$\|R(A(S)) - R^* - \Delta_{\mathrm{opt}}\|_q \leq C\left( q\gamma_q \log(N) + \sqrt{\frac{qB(\|R(A(S)) - R^* - \Delta_{\mathrm{opt}}\|_q + \mathbb{E}[\Delta_{\mathrm{opt}}])}{N}} + \frac{qM}{N}\right).$$

By invoking Lemma 5 to the above self-bounding inequality with $x = \|R(A(S)) - R^* - \Delta_{\mathrm{opt}}\|_q$, $a = C(q\gamma_q \log(N) + \frac{qM}{N})$, $b = \frac{qB}{N}$, and $c = \mathbb{E}[\Delta_{\mathrm{opt}}]$ we immediately obtain that

$$\|R(A(S)) - R^* - \Delta_{\mathrm{opt}}\|_q \lesssim \mathbb{E}[\Delta_{\mathrm{opt}}] + q\gamma_q \log(N) + \frac{(M+B)q}{N}.$$

This gives the desired bound in part (a).

*Part (b)*: Under the given conditions in part (b), we can bound the term $T$ in Eq. (18) as follows for $q \geq 2$:

---

[2]It is possible to relax the $M$-boundedness condition on the loss function $\ell$ to its sub-Gaussian or sub-exponential counterparts by alternatively applying general Bernstein-type inequalities for sums of independent sub-Gaussian or sub-exponential random variables (Vershynin, 2018) in this part of proof. For the sake of simplicity and transparency of exposition, here we choose to work on the bounded loss while keeping in mind that the requirement is not essential.

$$T = \left\| \frac{1}{N} \sum_{i=1}^{N} \mathbb{E}\left[\ell(w^*(S'); Z_i) - \ell(A(S'); Z_i) \mid Z_i\right] - (R^* - \mathbb{E}[R(A(S'))]) \right\|_q$$

$$\overset{\zeta_1}{\leq} \frac{2\sqrt{2\kappa q}}{N} \sqrt{\left\| \sum_{i=1}^{N} \left(\mathbb{E}\left[\ell(w^*(S'); Z_i) - \ell(A(S'); Z_i) \mid Z_i\right] - (R^* - \mathbb{E}[R(A(S'))])\right)^2 \right\|_{q/2}}$$

$$\leq \frac{4\sqrt{\kappa q}}{N} \sqrt{\sum_{i=1}^{N} \left\| \mathbb{E}^2\left[\ell(w^*(S'); Z_i) - \ell(A(S'); Z_i) \mid Z_i\right] + (R^* - \mathbb{E}[R(A(S'))])^2 \right\|_{q/2}}$$

$$\leq \frac{4\sqrt{\kappa q}}{N} \sqrt{\sum_{i=1}^{N} \left\| \mathbb{E}^2\left[\ell(w^*(S'); Z_i) - \ell(A(S'); Z_i) \mid Z_i\right] \right\|_{q/2} + N \mathbb{E}^2[R(w^*(S')) - R(A(S'))]}$$

$$\overset{\zeta_2}{\leq} \frac{4\sqrt{\kappa q}}{N} \sqrt{\sum_{i=1}^{N} \left\| \mathbb{E}^2\left[G\|w^*(S') - A(S')\| \mid Z_i\right] \right\|_{q/2} + N \mathbb{E}^2\left[G\|w^*(S') - A(S')\|\right]}$$

$$\overset{\zeta_3}{\leq} \frac{4G\sqrt{2\kappa q}}{\sqrt{N}} \sqrt{\mathbb{E}\left[\|w^*(S') - A(S')\|^2\right]}$$

$$\overset{\zeta_4}{\leq} \frac{8G\sqrt{\kappa q}}{\sqrt{N}} \sqrt{\frac{1}{\mu}\mathbb{E}\left[R(A(S')) - R^*\right]} = \frac{8G\sqrt{\kappa q}}{\sqrt{N\mu}} \sqrt{\mathbb{E}\left[R(A(S')) - R^* - \Delta_{\text{opt}}\right] + \mathbb{E}[\Delta_{\text{opt}}]}$$

$$\leq \frac{8G\sqrt{\kappa q}}{\sqrt{N\mu}} \sqrt{\left\| R(A(S)) - R^* - \Delta_{\text{opt}} \right\|_q + \mathbb{E}[\Delta_{\text{opt}}]}$$

where in "$\zeta_1$" we have used Proposition 2, in "$\zeta_2$" we have used the Lipschitz-loss condition, in "$\zeta_3$" we have used $\left\| \mathbb{E}^2\left[G\|w^*(S') - A(S')\| \mid Z_i\right] \right\|_{q/2} = \mathbb{E}^2\left[G\|w^*(S') - A(S')\|\right] \leq G^2 \mathbb{E}\left[\|w^*(S') - A(S')\|^2\right]$, in "$\zeta_4$" we have used Assumption 2, and the last inequality is due to Jensen's inequality. Then, plugging the above bound of term $T$ into Eq. (18) yields that for some universal constant $C$:

$$\|R(A(S)) - R^* - \Delta_{\text{opt}}\|_q \leq C\left(q\gamma_q \log(N) + G\sqrt{\frac{q}{N\mu}}\sqrt{\|R(A(S)) - R^* - \Delta_{\text{opt}}\|_q + \mathbb{E}[\Delta_{\text{opt}}]}\right).$$

Invoking Lemma 5 to the above inequality with $x = \|R(A(S)) - R^* - \Delta_{\text{opt}}\|_q$, $a = Cq\gamma_q \log(N)$, $b = \frac{qG^2}{\mu N}$, and $c = \mathbb{E}[\Delta_{\text{opt}}]$ yields

$$\|R(A(S)) - R^* - \Delta_{\text{opt}}\|_q \lesssim \mathbb{E}[\Delta_{\text{opt}}] + q\gamma_q \log(N) + \frac{qG^2}{\mu N}.$$

This gives the desired bound in part (b). The proof is completed. $\square$

## C  PROOFS FOR SECTION 3

### C.1  PROOF OF LEMMA 1

*Proof.* Let us consider the following event about the restricted strong convexity of $R_S$:

$$\mathcal{E} : R_S \text{ is } \mu_k\text{-strongly convex.}$$

Let $Y = 1_{\mathcal{E}}$ be the indication random variable associated with $\mathcal{E}$. Then by Assumption 4 we have $\mathbb{P}(Y = 1) \geq 1 - \delta_N$. Suppose that $\mathcal{E}$ occurs such that $Y = 1$. Then,

$$R_S(w^*_{S^{(i)}|J}) - R_S(w^*_{S|J})$$

$$= \frac{1}{N} \sum_{j \neq i} \left( \ell(w^*_{S^{(i)}|J}; Z_j) - \ell(w^*_{S|J}; Z_j) \right) + \frac{1}{N} \left( \ell(w^*_{S^{(i)}|J}; Z_i) - \ell(w^*_{S|J}; Z_i) \right)$$

$$= R_{S^{(i)}}(w^*_{S^{(i)}|J}) - R_{S^{(i)}}(w^*_{S|J}) + \frac{1}{N} \left( \ell(w^*_{S^{(i)}|J}; Z_i) - \ell(w^*_{S|J}; Z_i) \right)$$

$$\quad - \frac{1}{N} \left( \ell(w^*_{S^{(i)}|J}; Z'_i) - \ell(w^*_{S|J}; Z'_i) \right)$$

$$\leq \frac{1}{N} \left| \ell(w^*_{S^{(i)}|J}; Z_i) - \ell(w^*_{S|J}; Z_i) \right| + \frac{1}{N} \left| \ell(w^*_{S^{(i)}|J}; Z'_i) - \ell(w^*_{S|J}; Z'_i) \right|$$

$$\leq \frac{2G}{N} \left\| w^*_{S^{(i)}|J} - w^*_{S|J} \right\|,$$

where we have used the optimality of $w^*_{S^{(i)}|J}$ with respect to $R_{S^{(i)}}(w)$ and the Lipschitz continuity of loss. Since $\mathcal{E}$ occurs by assumption, $R_S$ is $\mu_k$-strongly convex. Since $w^*_{S|J}$ is optimal for $R_S(w)$ over the supporting set $J$, we have

$$R_S(w^*_{S^{(i)}|J}) \geq R_S(w^*_{S|J}) + \frac{\mu_k}{2} \left\| w^*_{S^{(i)}|J} - w^*_{S|J} \right\|^2.$$

Combing the preceding two inequalities yields $\left\| w^*_{S^{(i)}|J} - w^*_{S|J} \right\| \leq \frac{4G}{\mu_k N}$. Consequently from the Lipschitz continuity of $\ell$ we have that for any $Z \in \mathcal{Z}$, the following holds conditioned on $Y = 1$:

$$\left| \ell(w^*_{S^{(i)}|J}; Z) - \ell(w^*_{S|J}; Z) \right| \leq G \left\| w^*_{S^{(i)}|J} - w^*_{S|J} \right\| \leq \frac{4G^2}{\mu_k N}. \tag{19}$$

In the complementary case of $Y = 0$, in view of Assumption 5, it always holds that

$$|\ell(w^*_{S^{(i)}|J}; Z) - \ell(w^*_{S|J}; Z)| \leq G \left\| w^*_{S^{(i)}|J} - w^*_{S|J} \right\| \leq 2GD. \tag{20}$$

Let us consider $q_u := \frac{\log\left(\frac{1}{\delta_N}\right)}{\log(N)}$. By assumption $q_u \geq 2$. Then for $2 \leq q \leq q_u$, it can be verified that

$$\mathbb{E}\left[ \left| \ell(w^*_{S^{(i)}|J}; Z) - \ell(w^*_{S|J}; Z) \right|^q \right]$$

$$= \mathbb{P}(Y = 1)\mathbb{E}\left[ \left| \ell(w^*_{S^{(i)}|J}; Z) - \ell(w^*_{S|J}; Z) \right|^q \mid Y = 1 \right]$$

$$\quad + \mathbb{P}(Y = 0)\mathbb{E}\left[ \left| \ell(w^*_{S^{(i)}|J}; Z) - \ell(w^*_{S|J}; Z) \right|^q \mid Y = 0 \right]$$

$$\leq \left( \frac{4G^2}{\mu_k N} \right)^q + \delta_N(2GD)^q = \left( \frac{4G^2}{\mu_k N} \right)^q + \frac{1}{N^{q_u}}(2GD)^q \leq \left( \frac{4G^2}{\mu_k N} \right)^q + \frac{1}{N^q}(2GD)^q.$$

It follows that for all $2 \leq q \leq q_u$

$$\left\| \ell(w^*_{S^{(i)}|J}; Z) - \ell(w^*_{S|J}; Z) \right\|_q \leq \left( \left( \frac{4G^2}{\mu_k N} \right)^q + \frac{1}{N^q}(2GD)^q \right)^{1/q} \leq \frac{1}{N} \left( \frac{4G^2}{\mu_k} + 2GD \right),$$

where we have used $a^q + b^q \leq (a+b)^q$ for $a, b > 0$ and $q \geq 2$. For the complementary case $q > q_u$, it is trivial to show that

$$\left\| \ell(w^*_{S^{(i)}|J}; Z) - \ell(w^*_{S|J}; Z) \right\|_q \leq 2GD \leq \frac{2GD}{q_u}q.$$

Assembling the preceding two bounds yields the desired $L_q$-stability bound. $\qquad \square$

## C.2 PROOF OF THEOREM 4

Let us denote $a_+ = \max\{a, 0\}$. We need the following key result in our analysis, whose proof idea draws large inspiration from that of Theorem 3 with proper modifications for handing the challenges imposed by the combinatorial optimization nature of $L_0$-ERM.

**Lemma 7.** *Suppose that Assumptions 3, 4, 5 hold. Assume that* $\frac{\log(1/\delta_N)}{\log(N)} \geq 2$. *Then for any* $\delta \in (0, e^{-1})$, *it holds with probability at least* $1 - \delta$ *that*

$$
\sup_{J \subseteq [d], |J|=k} R(w^*_{S|J}) - R(w^*_{\bar{k}})
$$
$$
\lesssim \frac{GD \left(k \log\left(\frac{ed}{k}\right) + \log\left(\frac{e}{\delta}\right)\right)^2 \log^2(N)}{\log(1/\delta_N)} + \left(\log(N)\left(\frac{G^2}{\mu_k} + GD\right) + \frac{G^2}{\mu}\right) \frac{k \log\left(\frac{ed}{k}\right) + \log\left(\frac{e}{\delta}\right)}{N}
$$
$$
+ G\sqrt{\frac{\left(k \log\left(\frac{ed}{k}\right) + \log\left(\frac{e}{\delta}\right)\right)\left(R(w^*_{\bar{k}}) - R(w^*)\right)}{N\mu}} + \left(R_S(w^*_{S|J}) - R_S(w^*_{S,\bar{k}})\right)_+
$$
$$
+ \mathbb{E}\left[\left(R_S(w^*_{S|J}) - R_S(w^*_{S,\bar{k}})\right)_+\right].
$$

*Proof.* Given a fixed index set $J \subseteq [d]$ with $|J| = k$, we can show that the following holds for any $q \geq 1$:

$$
\left\| R(w^*_{S|J}) - R_S(w^*_{S|J}) + R_S(w^*) - R(w^*) \right\|_q
$$
$$
= \left\| \Gamma_{S|J} + \mathbb{E}[R(w^*_{S|J})] - \frac{1}{N}\sum_{i=1}^N \mathbb{E}[\ell(w^*_{S'|J}; Z_i) \mid Z_i] + R_S(w^*) - R(w^*) \right\|_q \tag{21}
$$
$$
\leq \left\| \Gamma_{S|J} \right\|_q + \underbrace{\left\| \mathbb{E}[R(w^*_{S|J})] - \frac{1}{N}\sum_{i=1}^N \mathbb{E}[\ell(w^*_{S'|J}; Z_i) \mid Z_i] - R(w^*) + R_S(w^*) \right\|_q}_{T},
$$

where

$$
\Gamma_{S|J} = R(w^*_{S|J}) - R_S(w^*_{S|J}) - \mathbb{E}[R(w^*_{S|J})] + \frac{1}{N}\sum_{i=1}^N \mathbb{E}[\ell(w^*_{S'|J}; Z_i) \mid Z_i].
$$

In view of Lemma 1 we have that $w^*_{S|J}$ has $L_q$-stability by

$$
\gamma_q = \frac{1}{N}\left(\frac{4G^2}{\mu_k} + 2GD\right) + \frac{2GD\log(N)q}{\log(1/\delta_N)}.
$$

Then invoking Lemma 6 over the supporting set $J$ yields

$$
\left\| \Gamma_{S|J} \right\|_q \lesssim q\gamma_q \log(N) = \frac{q\log(N)}{N}\left(\frac{4G^2}{\mu_k} + 2GD\right) + \frac{2GD\log^2(N)q^2}{\log(1/\delta_N)}. \tag{22}
$$

We now bound the term $T$ in Eq. (21) as follows:

$$
T = \left\| \frac{1}{N} \sum_{i=1}^{N} \mathbb{E}\left[ w^*; Z_i \right) - \ell(w^*_{S'|J}; Z_i) \mid Z_i \right] - (R(w^*) - \mathbb{E}[R(w^*_{S'|J})]) \right\|_q
$$

$$
\overset{\zeta_1}{\leq} \frac{2\sqrt{2\kappa q}}{N} \sqrt{ \left\| \sum_{i=1}^{N} \left( \mathbb{E}\left[ \ell(w^*; Z_i) - \ell(w^*_{S'|J}; Z_i) \mid Z_i \right] - (R(w^*) - \mathbb{E}[R(w^*_{S'|J})]) \right)^2 \right\|_{q/2} }
$$

$$
\leq \frac{4\sqrt{\kappa q}}{N} \sqrt{ \sum_{i=1}^{N} \left\| \mathbb{E}^2\left[ \ell(w^*; Z_i) - \ell(w^*_{S'|J}; Z_i) \mid Z_i \right] + \left( R(w^*) - \mathbb{E}[R(w^*_{S'|J})] \right)^2 \right\|_{q/2} }
$$

$$
\leq \frac{4\sqrt{\kappa p}}{N} \sqrt{ \sum_{i=1}^{N} \left\| \mathbb{E}^2\left[ \ell(w^*; Z_i) - \ell(w^*_{S'|J}; Z_i) \mid Z_i \right] \right\|_{q/2} + N\mathbb{E}^2\left[ R(w^*) - R(w^*_{S'|J}) \right] } \quad (23)
$$

$$
\overset{\zeta_2}{\leq} \frac{4\sqrt{\kappa p}}{N} \sqrt{ \sum_{i=1}^{N} \left\| \mathbb{E}^2\left[ G \left\| w^* - w^*_{S'|J} \right\| \mid Z_i \right] \right\|_{q/2} + N\mathbb{E}^2\left[ G \left\| w^* - w^*_{S'|J} \right\| \right] }
$$

$$
\overset{\zeta_3}{\leq} \frac{4G\sqrt{2\kappa p}}{\sqrt{N}} \sqrt{ \mathbb{E}\left[ \left\| w^* - w^*_{S'|J} \right\|^2 \right] }
$$

$$
\overset{\zeta_4}{\leq} \frac{8G\sqrt{\kappa q}}{\sqrt{N}} \sqrt{ \frac{1}{\mu} \mathbb{E}\left[ R(w^*_{S'|J}) - R(w^*) \right] } = \frac{8G\sqrt{\kappa q}}{\sqrt{N\mu}} \sqrt{ \mathbb{E}\left[ R(w^*_{S|J}) - R(w^*) \right] },
$$

where in "$\zeta_1$" we have used Proposition 2, in "$\zeta_2$" we have used the Lipschitz-loss assumption, in "$\zeta_3$" we have used $\left\| \mathbb{E}^2\left[ G\|w^* - w^*_{S'|J}\| \mid Z_i \right] \right\|_{q/2} = \mathbb{E}^2\left[ G\|w^* - w^*_{S'|J}\| \right] \leq G^2\mathbb{E}\left[ \|w^* - w^*_{S'|J}\|^2 \right]$, in "$\zeta_4$" we have used the strong convexity condition in Assumption 4. Plugging Eq. (22) and Eq. (23) into Eq. (21) yields that for any $q \geq 2$,

$$
\left\| R(w^*_{S|J}) - R(w^*) - R_S(w^*_{S|J}) + R_S(w^*) \right\|_q
$$

$$
\leq \frac{q\log(N)}{N} \left( \frac{4G^2}{\mu_k} + 2GD \right) + \frac{2q^2 GD \log^2(N)}{\log(1/\delta_N)} + 8G\sqrt{ \frac{\kappa q \mathbb{E}\left[ R(w^*_{S|J}) - R(w^*) \right]}{N\mu} }. \quad (24)
$$

Next we need to upper bound the factor $\mathbb{E}\left[ R(w^*_{S|J}) - R(w^*) \right]$ in the second term of the above bound. To do so, let us consider $q = 2$ in Eq. (24). It follows from the optimality of $w^*$ and $w^*_{S|J}$ that

$$
\mathbb{E}\left[ R(w^*_{S|J}) - R(w^*) - R_S(w^*_{S|J}) + R_S(w^*) \right]
$$

$$
\leq \left\| R(w^*_{S|J}) - R(w^*) - R_S(w^*_{S|J}) + R_S(w^*) \right\|_2
$$

$$
\overset{Eq.\ (24)}{\leq} \frac{\log(N)}{N} \left( \frac{8G^2}{\mu_k} + 4GD \right) + \frac{8GD \log^2(N)}{\log(1/\delta_N)} + 8G\sqrt{ \frac{2\kappa \mathbb{E}\left[ R(w^*_{S|J}) - R(w^*) \right]}{N\mu} }
$$

$$
\leq \frac{\log(N)}{N} \left( \frac{8G^2}{\mu_k} + 4GD \right) + \frac{8GD \log^2(N)}{\log(1/\delta_N)} + \frac{\mathbb{E}\left[ R(w^*_{S|J}) - R(w^*) \right]}{2} + \frac{64\kappa G^2}{N\mu},
$$

where in the first inequality we have used Cauchy-Schwarz inequality, and in the last inequality we have used the fact $\sqrt{ab} \leq \frac{a}{2t} + \frac{bt}{2}$ for any $a, b, t > 0$. Rearranging both sides of the above inequality

with simple algebra leads to

$$
\begin{aligned}
&\mathbb{E}\left[R(w_{S|J}^*) - R(w^*)\right] \\
&\leq 2\mathbb{E}\left[R_S(w_{S|J}^*) - R_S(w^*)\right] + \frac{\log(N)}{N}\left(\frac{16G^2}{\mu_k} + 8GD\right) + \frac{16GD\log^2(N)}{\log(1/\delta_N)} + \frac{128\kappa G^2}{N\mu} \\
&= 2\mathbb{E}\left[R_S(w_{S|J}^*) - R_S(w_{\bar{k}}^*) + R_S(w_{\bar{k}}^*) - R_S(w^*)\right] \\
&\quad + \frac{\log(N)}{N}\left(\frac{16G^2}{\mu_k} + 8GD\right) + \frac{16GD\log^2(N)}{\log(1/\delta_N)} + \frac{128\kappa G^2}{N\mu} \\
&= 2\mathbb{E}\left[R_S(w_{S|J}^*) - R_S(w_{\bar{k}}^*)\right] + 2(R(w_{\bar{k}}^*) - R(w^*)) \\
&\quad + \frac{\log(N)}{N}\left(\frac{16G^2}{\mu_k} + 8GD\right) + \frac{16GD\log^2(N)}{\log(1/\delta_N)} + \frac{128\kappa G^2}{N\mu} \\
&\leq 2\mathbb{E}\left[R_S(w_{S|J}^*) - R_S(w_{S,\bar{k}}^*)\right] + 2(R(w_{\bar{k}}^*) - R(w^*)) \\
&\quad + \frac{\log(N)}{N}\left(\frac{16G^2}{\mu_k} + 8GD\right) + \frac{16GD\log^2(N)}{\log(1/\delta_N)} + \frac{128\kappa G^2}{N\mu} \\
&\leq \underbrace{2\mathbb{E}\left[\left(R_S(w_{S|J}^*) - R_S(w_{S,\bar{k}}^*)\right)_+\right]}_{T_1} + \underbrace{2\left(R(w_{\bar{k}}^*) - R(w^*)\right)}_{T_2} \\
&\quad + \underbrace{\frac{\log(N)}{N}\left(\frac{16G^2}{\mu_k} + 8GD\right) + \frac{16GD\log^2(N)}{\log(1/\delta_N)} + \frac{128\kappa G^2}{N\mu}}_{T_3}.
\end{aligned}
\tag{25}
$$

Plugging Eq. (25) into Eq. (24) yields that for any $q \geq 2$

$$
\begin{aligned}
&\left\|R(w_{S|J}^*) - R(w^*) - R_S(w_{S|J}^*) + R_S(w^*)\right\|_q \\
&\leq \frac{q\log(N)}{N}\left(\frac{4G^2}{\mu_k} + 2GD\right) + \frac{2GDq^2\log^2(N)}{\log(1/\delta_N)} + 8G\sqrt{\frac{\kappa q(T_1 + T_2 + T_3)}{N\mu}} \\
&\leq \frac{q\log(N)}{N}\left(\frac{4G^2}{\mu_k} + 2GD\right) + \frac{2GDq^2\log^2(N)}{\log(1/\delta_N)} + 8G\sqrt{\frac{\kappa q(T_1 + T_3)}{N\mu}} + 8G\sqrt{\frac{\kappa q T_2}{N\mu}} \\
&\overset{\varsigma_1}{\leq} \frac{q\log(N)}{N}\left(\frac{4G^2}{\mu_k} + 2GD\right) + \frac{2GDq^2\log^2(N)}{\log(1/\delta_N)} + 8G\sqrt{\frac{\kappa q T_2}{N\mu}} + T_1 + T_3 + \frac{16\kappa q G^2}{N\mu} \\
&\lesssim \frac{q\log(N)}{N}\left(\frac{G^2}{\mu_k} + GD\right) + \frac{GDq^2\log^2(N)}{\log(1/\delta_N)} + \frac{qG^2}{N\mu} + \mathbb{E}\left[\left(R_S(w_{S|J}^*) - R_S(w_{S,\bar{k}}^*)\right)_+\right] \\
&\quad + G\sqrt{\frac{q\left(R(w_{\bar{k}}^*) - R(w^*)\right)}{N\mu}},
\end{aligned}
\tag{26}
$$

where in "$\varsigma_1$" we have again used the fact $\sqrt{ab} \leq \frac{a}{2t} + \frac{bt}{2}$ for $a, b, t > 0$. Now we are in the position to finally upper bound the desired sparse excess risk with respect to $w_{\bar{k}}^*$, which can be decomposed

in the following way:

$$
\begin{aligned}
R(w^*_{S|J}) - R(w^*_{\bar{k}}) =& R(w^*_{S|J}) - R(w^*) - R_S(w^*_{S|J}) + R_S(w^*) + R_S(w^*_{S|J}) - R_S(w^*_{\bar{k}}) \\
& + R_S(w^*_{\bar{k}}) - R_S(w^*) + R(w^*) - R(w^*_{\bar{k}}) \\
\leq& R(w^*_{S|J}) - R(w^*) - R_S(w^*_{S|J}) + R_S(w^*) + R_S(w^*_{S|J}) - R_S(w^*_{S,\bar{k}}) \\
& + R_S(w^*_{\bar{k}}) - R_S(w^*) + R(w^*) - R(w^*_{\bar{k}}) \\
\leq& \underbrace{R(w^*_{S|J}) - R(w^*) - R_S(w^*_{S|J}) + R_S(w^*)}_{T'_1} + \Big( R_S(w^*_{S|J}) - R_S(w^*_{S,\bar{k}}) \Big)_+ \\
& + \underbrace{R_S(w^*_{\bar{k}}) - R_S(w^*) + R(w^*) - R(w^*_{\bar{k}})}_{T'_2},
\end{aligned}
$$

$$(27)$$

where in the first inequality we have used the fact $R_S(w^*_{S,\bar{k}}) \leq R_S(w^*_{\bar{k}})$. We are going to bound the above two terms $T'_1$ and $T'_2$ respectively with high probability. Concerning $T'_1$, since Eq. (26) holds for all $q \geq 2$, in view of the second part of Lemma 4 (with $q_l = 2$ and $q_u = \infty$) we can show that the following holds with probability at least $1 - \frac{\delta}{2}$ for any $\delta \in (0, e^{-1})$:

$$
\begin{aligned}
T'_1 =& R(w^*_{S|J}) - R(w^*) - R_S(w^*_{S|J}) + R_S(w^*) \\
\leq& \left| R(w^*_{S|J}) - R(w^*) - R_S(w^*_{S|J}) + R_S(w^*) \right| \\
\lesssim& \frac{GD \log^2\left(\frac{e}{\delta}\right) \log^2(N)}{\log(1/\delta_N)} + \left( \log(N) \left( \frac{G^2}{\mu_k} + GD \right) + \frac{G^2}{\mu} \right) \frac{\log\left(\frac{e}{\delta}\right)}{N} \\
& + G\sqrt{\frac{\log\left(\frac{e}{\delta}\right) \left( R(w^*_{\bar{k}}) - R(w^*) \right)}{N\mu}} + \mathbb{E}\left[ \Big( R_S(w^*_{S|J}) - R_S(w^*_{S,\bar{k}}) \Big)_+ \right].
\end{aligned}
$$

$$(28)$$

Regarding the term $T'_2$, similar to the argument of Eq. (23), we can bound its $L_q$-norm for any $q \geq 2$ as follows:

$$
\begin{aligned}
\|T'_2\|_q =& \left\| \frac{1}{N} \sum_{i=1}^N \left( \ell(w^*_{\bar{k}}; Z_i) - \ell(w^*; Z_i) \right) - \left( R(w^*_{\bar{k}}) - R(w^*) \right) \right\|_q \\
\leq& \frac{2\sqrt{2\kappa q}}{N} \sqrt{\left\| \sum_{i=1}^N \left( \ell(w^*_{\bar{k}}; Z_i) - \ell(w^*; Z_i) - \left( R(w^*_{\bar{k}}) - R(w^*) \right) \right)^2 \right\|_{q/2}} \\
\leq& \frac{4\sqrt{\kappa q}}{N} \sqrt{\sum_{i=1}^N \left\| \left( \ell(w^*_{\bar{k}}; Z_i) - \ell(w^*; Z_i) \right)^2 + \left( R(w^*_{\bar{k}}) - R(w^*) \right)^2 \right\|_{q/2}} \\
\leq& \frac{4\sqrt{\kappa q}}{N} \sqrt{\sum_{i=1}^N \left\| \left( \ell(w^*_{\bar{k}}; Z_i) - \ell(w^*; Z_i) \right)^2 \right\|_{q/2} + N \left( R(w^*_{\bar{k}}) - R(w^*) \right)^2} \\
\leq& \frac{4\sqrt{2\kappa q}}{N} \sqrt{NG^2 \left\| w^*_{\bar{k}} - w^* \right\|^2} \\
\leq& \frac{8G\sqrt{\kappa q}}{\sqrt{N\mu}} \sqrt{R(w^*_{\bar{k}}) - R(w^*)}.
\end{aligned}
$$

By invoking the second part of Lemma 4 with $q_l = 2$ and $q_u = \infty$ we can translate the above moment bound into the following exponential tail bound that holds with probability at least $1 - \frac{\delta}{2}$ for any $\delta \in (0, e^{-1})$:

$$
T'_2 \lesssim G\sqrt{\frac{\log\left(\frac{e}{\delta}\right) \left( R(w^*_{\bar{k}}) - R(w^*) \right)}{N\mu}}.
$$

$$(29)$$

By plugging Eq. (28) and Eq. (29) into Eq. (27) and applying union probability argument we obtain that the following holds with probability at least $1 - \delta$ for any $\delta \in (0, e^{-1})$:

$$R(w^*_{S|J}) - R(w^*_{\bar{k}})$$

$$\lesssim \frac{GD \log^2\left(\frac{e}{\delta}\right) \log^2(N)}{\log(1/\delta_N)} + \left(\log(N)\left(\frac{G^2}{\mu_k} + GD\right) + \frac{G^2}{\mu}\right)\frac{\log\left(\frac{e}{\delta}\right)}{N} + G\sqrt{\frac{\log\left(\frac{e}{\delta}\right)\left(R(w^*_{\bar{k}}) - R(w^*)\right)}{N\mu}}$$

$$+ \left(R_S(w^*_{S|J}) - R_S(w^*_{S,\bar{k}})\right)_+ + \mathbb{E}\left[\left(R_S(w^*_{S|J}) - R_S(w^*_{S,\bar{k}})\right)_+\right].$$

This proves the desired sparse excess risk bound over the supporting set $J$.

As the final step, since there are at most $\binom{d}{k} \leq \left(\frac{ed}{k}\right)^k$ different $J$ with $|J| = k$, by union probability we can show that the following holds with probability $1 - \delta$ for any $\delta \in (0, e^{-1})$:

$$\sup_{J \subseteq [d], |J| = k} R(w^*_{S|J}) - R(w^*_{\bar{k}})$$

$$\lesssim \frac{GD\left(k\log\left(\frac{ed}{k}\right) + \log\left(\frac{e}{\delta}\right)\right)^2 \log^2(N)}{\log(1/\delta_N)} + \left(\log(N)\left(\frac{G^2}{\mu_k} + GD\right) + \frac{G^2}{\mu}\right)\frac{k\log\left(\frac{ed}{k}\right) + \log\left(\frac{e}{\delta}\right)}{N}$$

$$+ G\sqrt{\frac{\left(k\log\left(\frac{ed}{k}\right) + \log\left(\frac{e}{\delta}\right)\right)\left(R(w^*_{\bar{k}}) - R(w^*)\right)}{N\mu}} + \left(R_S(w^*_{S|J}) - R_S(w^*_{S,\bar{k}})\right)_+$$

$$+ \mathbb{E}\left[\left(R_S(w^*_{S|J}) - R_S(w^*_{S,\bar{k}})\right)_+\right].$$

This completes the proof. $\qquad\square$

Now we are in the position to prove the main result in Theorem 4.

*Proof of Theorem 4.* Let us consider $\tilde{J} = \text{supp}(\tilde{w}_{S,k})$. By definition we have $\tilde{w}_{S,k} = w^*_{S|\tilde{J}}$. Then invoking Lemma 7 yields that for any $\delta \in (0, e^{-1})$, the following holds with probability at least $1 - \delta$:

$$R(\tilde{w}_{S,k}) - R(w^*_{\bar{k}}) = R(w^*_{S|\tilde{J}}) - R(w^*_{\bar{k}})$$

$$\lesssim \frac{GD\left(k\log\left(\frac{ed}{k}\right) + \log\left(\frac{e}{\delta}\right)\right)^2 \log^2(N)}{\log(1/\delta_N)} + \left(\log(N)\left(\frac{G^2}{\mu_k} + GD\right) + \frac{G^2}{\mu}\right)\frac{k\log\left(\frac{ed}{k}\right) + \log\left(\frac{e}{\delta}\right)}{N}$$

$$+ G\sqrt{\frac{\left(k\log\left(\frac{ed}{k}\right) + \log\left(\frac{e}{\delta}\right)\right)\left(R(w^*_{\bar{k}}) - R(w^*)\right)}{N\mu}} + \left(R_S(w^*_{S|\tilde{J}}) - R_S(w^*_{S,\bar{k}})\right)_+$$

$$+ \mathbb{E}\left[\left(R_S(w^*_{S|\tilde{J}}) - R_S(w^*_{S,\bar{k}})\right)_+\right].$$

$$\lesssim \frac{GD\left(k\log\left(\frac{ed}{k}\right) + \log\left(\frac{e}{\delta}\right)\right)^2 \log^2(N)}{\log(1/\delta_N)} + \left(\log(N)\left(\frac{G^2}{\mu_k} + GD\right) + \frac{G^2}{\mu}\right)\frac{k\log\left(\frac{ed}{k}\right) + \log\left(\frac{e}{\delta}\right)}{N}$$

$$+ G\sqrt{\frac{\left(k\log\left(\frac{ed}{k}\right) + \log\left(\frac{e}{\delta}\right)\right)\left(R(w^*_{\bar{k}}) - R(w^*)\right)}{N\mu}} + \Delta_{\bar{k},\text{opt}} + \mathbb{E}\left[\Delta_{\bar{k},\text{opt}}\right].$$

This proves the desired sparse excess risk bound. $\qquad\square$

## D  OTHER RELATED WORK

**Uniform stability and exponential generalization.** Stimulated by a recent landmark work of Hardt et al. (2016), there is a renewed interest in the use of uniform stability for deriving generalization bounds for various learning algorithms and paradigms including stochastic gradient descent (SGD) (Kuzborskij & Lampert, 2018; Charles & Papailiopoulos, 2018; Lei & Ying, 2020), stochastic model based optimization (Wang et al., 2017; Deng & Gao, 2021), optimization based meta learning (Zhou et al., 2019), and differential privacy stochastic optimization (Bassily et al., 2019; Feldman et al., 2020). Compared to other stability arguments, uniform stability is notorious for implying high-probability generalization bounds in addition to the traditional in-expectation bounds. Until very recently, the basic result of Bousquet & Elisseeff (2002), as expressed in Eq. (2), remains the best known exponential generalization bound for uniformly stable algorithms. Using some elegant proof techniques from adaptive data analysis, Feldman & Vondrák (2018) managed to replace the first term in Eq. (2) by $\sqrt{\gamma_u \log(\frac{1}{\delta})}$, which leads to an improvement whenever $\gamma_u \gtrsim \frac{1}{N}$. Soon after, a series of breakthrough results were obtained (Feldman & Vondrák, 2019; Bousquet et al., 2020) using tighter concentration bounds for sum of random functions, which eventually improve the stability dependent rate to a near-optimal one $\gamma_u \log(N) \log\left(\frac{1}{\delta}\right)$ as shown in Eq. (4). Additionally under the generalized Bernstein condition, these state-of-the-art results lead to $\mathcal{O}(\frac{1}{N})$ excess risk bounds for uniformly stable algorithms (Klochkov & Zhivotovskiy, 2021).

**Non-uniform stability and exponential generalization.** More broadly for non-uniformly stable algorithms, exponential generalization bounds have also been shown to be possible under various weaker and distribution-dependent notions of stability. As an early work in this line, Kutin & Niyogi (2002) showed that under the so called "almost-everywhere" stability, which is a high-probability counterpart of uniform stability, generalization bounds that hold with overwhelming probability are still possible in view of certain modified McDiarmid's inequality (Kutin, 2002). Later, Rakhlin et al. (2005) revisited the bounded-difference results of Kutin & Niyogi (2002) in a more straightforward manner by using a powerful moment extension of Efron-Stein inequality (Boucheron et al., 2005). Recently for general $L_q$-stable algorithms, the exponential leave-one-out generalization bounds were derived using moment or exponential extensions of Efron-Stein inequality, with applications found in ridge regression, $k$-nearest neighbor classification and $k$-folds cross-validation (Celisse & Guedj, 2016; Celisse & Mary-Huard, 2018; Abou-Moustafa & Szepesvári, 2019). However, when it comes to the recent break-through bounds of Feldman & Vondrák (2019); Bousquet et al. (2020), it is much less obvious how to easily extend these near-optimal bounds under the almost-everywhere stability via simply incorporating the low probability failure events into concentration inequality. This is actually in sharp contrast to what have been done by Feldman & Vondrák (2019, Theorem 4.5) and Bassily et al. (2020, Theorem 2.1) for stochastic learning algorithms with uniform stability (over the randomness of data) holding with high probability over the internal randomness of algorithm. We refer the interested readers to Boucheron et al. (2013); Kontorovich (2014); Combes (2015); Warnke (2016); Maurer & Pontil (2021) for more results on concentration inequalities beyond bounded-difference conditions, which are fundamental for deriving exponential bounds for non-uniformly stable algorithms.

**Generalization analysis of $\ell_0$-estimators.** We further briefly review some prior results on the generalization guarantees for statistical learning under cardinality constraints, which is the theme of the application part of our work. For the $\ell_0$-ERM estimator as expressed in Eq. (8), provided that the solution is exactly known, a series of uniform excess risk bounds were derived over binary prediction classes (Chen & Lee, 2018; 2020) and bounded liner prediction classes (Foster & Syrgkanis, 2019). The exact solutions of $\ell_0$-ERM, however, is computationally intractable in general high-dimensional cases due to the NP-hardness of problem. Therefore, it is more realistic and desirable to establish generalization bounds for approximate $\ell_0$-estimators such as the IHT-style algorithms (Yuan et al., 2018; Garg & Khandekar, 2009; Li et al., 2016). Particularly for misspecified sparsity models, a set of sparse excess risk bounds with slow and fast rates were established through the lens of uniform stability theory under proper regularity conditions (Yuan & Li, 2022).

