# OpenReview forum: "Exponential Generalization Bounds with Near-Optimal Rates for $L_q$-Stable Algorithms"
_ICLR.cc/2023/Conference — ICLR 2023 poster_

### Official Review · Reviewer_LaZo · 2022-10-20

**Confidence:** 3
**Correctness:** 4
**Technical Novelty And Significance:** 3
**Empirical Novelty And Significance:** Not applicable
**Recommendation:** 8

**Clarity, Quality, Novelty And Reproducibility:**

The background and significance of the result are both very well-explained in the introduction and throughout the paper. In general, the paper is well written, very clear, and easy to follow (even from someone like me, who works outside of this research area). Theorem 1 seems to be a result of general interest.

**Strength And Weaknesses:**

**Strength and Weaknesses**

The paper's technical and conceptual contributions are solid, clearly stated, and well-explained. The techniques seem sounds and the application to approximate $L_0$-ERM interesting.

**Questions/Feedback**

- Could you comment on the technical novelty of Theorem 1? I understand this is a generalization of another result, but where the bounded difference is distribution-dependent, so this question is more about the mechanics of the proof. In particular, in Appendix B.1, you state "*The proof is a generalization of that of Bousquet et al. (2020, Theorem 4) [...] We reproduce the proof here for the sake of completeness.*". Where and how does the proof differ from that of Bousquet et al. (2020, Theorem 4), and what is the main technical addition to the proof?
- In the statement of Theorem 1, the difference between the random variables $g_i(S)$ and $g_i(S^{(j)})$ is studied. On p.2, $S^{(j)}$ is defined as S but with the $j$-th sample point replaced by another sample point $Z_j'$. However the sample $S'=Z_1',\dots,Z_N'$ doesn't appear in the theorem statement. Is it simply that $S'$ is missing from the theorem statement? In that case, are S and S' sampled independently from the same distribution, as stated earlier on p.2?
- On p.3, it would be a good idea to at least explain in words what $R^*$ and $\Delta_{\text{opt}}$ are, instead of just referring to Eqn 8.
- Typo p.5: "substantially proves the overhead factor" -> "substantially improves the overhead factor"


**Summary Of The Paper:**

This papers studies the generalization properties of $L_q$-stable algorithms. This notion of stability is distribution-dependent, and thus less stringent analogue of uniform stability. The authors first derive a general inequality for the sum of functions of random variables with bounded difference (Theorem 1), and use the result to show generalization bounds improving the best known bounds for $L_q$-stable algorithms, going from a $\sqrt{N}$ to a $\log(N)$ factor in one of the terms, where $N$ is the sample size, which is important to avoid vacuous convergence rates in regimes where the uniform stability parameter is $\approx 1/\sqrt{N}$. They then use their result to show excess risk bounds for $L_0$-estimators.

**Summary Of The Review:**

I believe this is a solid contribution to the study of stable learning algorithms, and that the paper would be of general interest to the learning theory community.

---

> ### Author Response · Authors · 2022-11-16
> **Response to Reviewer LaZo**
>
> Thank you for your insightful review and appreciation of our work.
>
> > **Your comment:** Could you comment on the technical novelty of Theorem 1? ...Where and how does the proof differ from that of Bousquet et al. (2020, Theorem 4), and what is the main technical addition to the proof?
>
> **Our response:** The proof technique of Theorem 1 draws large inspiration from the strong arguments of Bousquet et al. (2020, Theorem 4), with slight extension from the uniform bounded difference setting to the $L_q$-norm bounded difference setting through using the generalized Efron-Stein inequality (Proposition 1) in places of McDiarmid’s inequality. While the techniques used in our analysis for this part are by no means ground-breaking, we believe that reviving the classic moments concentration inequalities along with the SOTA algorithmic stability theory for establishing near-tight generalization bounds for $L_q$-stable algorithms could be exciting and impactful.
>
> > **Your comment:** However the sample $S'=Z'_1,...,Z'_N$ doesn't appear in the theorem statement. Is it simply that $S'$ is missing from the theorem statement? In that case, are $S$ and $S'$ sampled independently from the same distribution, as stated earlier on p.2?
>
> **Our response:** It is true that $S'$ is an independent copy of $S$. We would like to clarify that the notation $S'$ has indeed been used in the statement and proof of Lemma 6 in Appendix B.3, which is fundamental for proving Theorem 3.
>
> > **Your comment:** On p.3, it would be a good idea to at least explain in words what $R^*$ and $\Delta_{opt}$ are, instead of just referring to Eqn 8.
>
> **Our response:** Per your suggestion, we have updated the exposition of notation by formally introducing the optimal population risk $R^*$ at the end of the *Problem setup* section on Page 2, and defining the sub-optimality $\Delta_{opt}$ right below Equation (5) where it is used for the first time. Thanks!
>
> > **Minor comments** on typos.
>
> **Our response:** Thanks for catching the typos which have been fixed in the revised draft.
>
> ## References:
>
> Olivier Bousquet, Yegor Klochkov, and Nikita Zhivotovskiy. Sharper bounds for uniformly stable
> algorithms. *Conference on Learning Theory*, pp. 610–626, 2020.

---

### Official Review · Reviewer_6gau · 2022-10-24

**Confidence:** 4
**Correctness:** 3
**Technical Novelty And Significance:** 3
**Empirical Novelty And Significance:** Not applicable
**Recommendation:** 6

**Clarity, Quality, Novelty And Reproducibility:**

This paper is well-structured. Their derived bounds are new. The key ingredient of their generalization bounds is a sharper concentration inequality, which is an extension of an existing moment inequality.

**Strength And Weaknesses:**

Strengths:
1. This paper derives near-optimal exponential generalization bounds for $L_q$-stable algorithms, which match the existing bounds of the uniformly stable algorithms.
2. Based on the notion of $L_q$-stability, they derive exponential risk bounds for computationally tractable sparsity estimation algorithms under mild assumptions.
3. To establish exponential generalization bounds for $L_q$-stable algorithms, they first extend the moment inequality in (Bousquet et al., 2022, Theorem 4) from under uniform bounded condition to $L_q$-norm bounded condition. The obtained concentration inequality can be applied to unbounded losses.

Weaknesses:
1. Theorem 2 is derived under $L_q$-norm boundness condition, which allows for learning with unbounded losses. Theorem 2 and Theorem 3 are both derived by applying the new moment inequality (Theorem 1). However, part (a) of theorem 3 requires the losses to be uniformly bounded. Compared to uniformly stability, $L_q$-stability is weaker since $L_q$-norm allows the losses to be unbounded. I am concerned about the incompatibility between the uniformly bounded assumption and the motivation of analysis of $L_q$-stability algorithm. Part (b) of theorem 3 assumes the loss to be Lipschitz with respect to the first argument. It is necessary to discuss the relationship between this Lipschitz assumption and bounded loss assumption.
2. Theorem 3 extends the near-optimal exponential risk bounds from uniform stable algorithms to $L_q$-stable algorithms. Theorem 4 presents the theoretical results on the sparse excess risk of the inexact $L_0$-ERM problems. It seems that Theorem 4 is an isolated application of $L_q$-stability, which is not related to previous theorems in the paper. I would recommend including an explanation of the relationship between Theorem 4 and previous theorems.
3. As pointed out in Remark 10, for misspecified sparsity models, their derived bound in Theorem 4 is slower than the existing results in (Yuan & Li, 2022, Theorem 3). They argue that their result is more broadly applicable. Besides the strong-signal condition, are there other assumptions that limit the application of results in (Yuan & Li, 2022, Theorem 3)? I would recommend including more discussions on differences of assumptions between Theorem 4 and results of (Yuan & Li, 2022, Theorem 3).

**Summary Of The Paper:**

This paper derives near-optimal exponential generalization bounds for $L_q$-stable algorithms. Compared to uniformly stability, $L_q$-stability is weaker and distribution dependent. To establish sharper generalization bounds, they first present the moment inequality for sum of random functions that satisfy the $L_q$-norm bounded difference condition, which is an extension of theorem 4 in (Bousquet et al., 2022). With this concentration inequality, they derive generalization bound and excess risk bound for $L_q$-stable algorithms. Moreover, they derive exponential risk bounds for computationally tractable sparsity estimation algorithms.

**Summary Of The Review:**

The paper derives some new exponential generalization bounds for $L_q$-stable algorithm. However, it is confusing for me to make bounded assumptions in the analysis of $L_q$-stable algorithms.

---

> ### Author Response · Authors · 2022-11-16
> **Response to Reviewer 6gau**
>
> Thank you for your insightful review and positive feedback. We sincerely hope that the main concerns raised in the review can be clarified by the following responses.
>
> > **Your comment:**  I am concerned about the incompatibility between the uniformly bounded assumption and the motivation of analysis of $L_q$-stability algorithm.
>
> **Our response:** We would like to clarify that in part (a) of Theorem 3, the $M$-bounded-loss condition is not essential and it can be relaxed to a sub-exponential (or sub-Gaussian) version by alternatively using the general Bernstein-type inequalities for sums of independent sub-exponential (or sub-Gaussian) random variables (Vershynin, 2018) in this part of proof. For the sake of simple and transparent exposition of results, here we choose to work with the uniformly bounded loss while keeping in mind that the analysis can be easily extended to the unbounded cases. Please see Remark 6 for the updated discussions regarding the bounded-loss condition in part (a) of Theorem 3.
>
> > **Your comment:** Part (b) of theorem 3 assumes the loss to be Lipschitz with respect to the first argument. It is necessary to discuss the relationship between this Lipschitz assumption and bounded loss assumption.
>
> **Our response:** Indeed, under the quadratic growth condition which is somewhat more stringent than the generalized Bernstein condition in part (a), the desired $\mathcal{O}(\frac{1}{N})$ fast rate of convergence is possible for Lipschitz-loss which is allowed to be arbitrarily unbounded. We have clarified this point for Theorem 3(b) in Remark 6 of the revised paper.
>
> >**Your comment:** It seems that Theorem 4 is an isolated application of $L_q$-stability, which is not related to previous theorems in the paper. I would recommend including an explanation of the relationship between Theorem 4 and previous theorems.
>
> **Our response:** Thanks for this insightful comment which we believe can be addressed by the following clarifications:
>
> 1. Concerning the connection between Theorem 4 and the main theorems in Section 2, we would like to stress that the proof technique of Theorem 4 follows closely that of Theorem 3. Indeed, the proof of Lemma 7, which is key to the proof of Theorem 4, uses identical ideas to that of Theorem 3 yet with some non-trivial extensions for handing the challenges imposed by the combinatorial optimization nature of $L_0$-ERM. In the revised draft, we have explicitly pointed out such a connection right above the statements of Theorem 4 and Lemma 7.
>
> 2. Also, we have further clarified that the inexact $L_0$-ERM application in Section 3 actually serves as an important motivation of our study on the generic $L_q$-stability and generalization theory. Please check the last paragraph above the *Notation* section on Page 2 for the related update.
>
>
> > **Your comment:** Besides the strong-signal condition, are there other assumptions that limit the application of results in (Yuan & Li, 2022, Theorem 3)? I would recommend including more discussions on differences of assumptions between Theorem 4 and results of (Yuan & Li, 2022, Theorem 3).
>
> **Our response:** Yes, in addition to the strong-signal condition, it is also assumed in the result of (Yuan & Li 2022, Theorem 3) that the loss should be bounded. In sharp contrast, our sparse excess risk bound in Theorem 4 is not relying on any strong-signal or bounded-loss conditions, and thus is exepected to have a broader range of applications in sparsity models. Please see Remark 10 for the updated discussions on the connections and differences between Theorem 4 and the related results of Yuan & Li (2022).
>
>
> ## References:
>
> Roman Vershynin. *High-dimensional probability: An introduction with applications in data science*, volume 47. Cambridge university press, 2018.
>
> Xiao-Tong Yuan and Ping Li. Stability and risk bounds of iterative hard thresholding. *IEEE Transactions on Information Theory*, 2022.

---

### Official Review · Reviewer_Ryqw · 2022-10-24

**Confidence:** 3
**Correctness:** 4
**Technical Novelty And Significance:** 4
**Empirical Novelty And Significance:** 4
**Recommendation:** 8

**Clarity, Quality, Novelty And Reproducibility:**


Nit: the definition (8) which is claimed to appear “shortly” on page 3 actually only appears on page 6. It would improve the exposition to define terms before usage.


**Strength And Weaknesses:**

The proposed results appear to both fill a natural hole in the literature and have reasonable applications. The exposition of prior work is well presented as well.

**Summary Of The Paper:**

This paper shows that Lq stable algorithms also generalize well with the “correct” rates. Specifically, recent analysis of uniformly stable algorithms has shown that uniform stability implies generalization at a $\tilde O(\gamma + 1/\sqrt{N})$ rate where $\gamma$ is the uniform stability parameter and $N$ is the sample size, improving prior results of $\tilde O(\gamma \sqrt{N} +  1/\sqrt{N})$. The current paper shows analogous improvement in results for Lq stable algorithms, where Lq stability indicates that the qth moment of difference of the losses on neighboring datasets is bounded. Since Lq stability is weaker than uniform stability, this applies to a wider variety of algorithms.



**Summary Of The Review:**

This result seems to fill an important gap in the literature on stability. The contribution appears strong and worthy of acceptance.

---

> ### Author Response · Authors · 2022-11-16
> **Response to Reviewer Ryqw**
>
> Thank you for your insightful review and positive evaluation of our work.
>
> > **Your comment:** The definition (8) which is claimed to appear “shortly” on page 3 actually only appears on page 6. It would improve the exposition to define terms before usage.
>
> **Our response:** We have followed your suggestion to introduce the concept of excess risk bound at the end of the *Problem setup* section on Page 2, and to define the sub-optimality $\Delta_{opt}$ right below Equation (5) where it first appears. Thanks!

---

### Official Review · Reviewer_dfYo · 2022-10-25

**Confidence:** 3
**Correctness:** 4
**Technical Novelty And Significance:** 4
**Empirical Novelty And Significance:** 4
**Recommendation:** 8

**Clarity, Quality, Novelty And Reproducibility:**

The paper is clear, well-written and easy to follow. To the best of the reviewer's knowledge, the paper is novel.

**Strength And Weaknesses:**

The paper significantly improves the previous result under the notion of $L_q$ stability, which is a novel and substantial contribution. The paper lacks numerical experiments to support sharpness of their bounds, but this is not a big deal for a theory paper.

**Summary Of The Paper:**

The paper derives near-optimal generalization bound under the notion of $L_q$-stability, extending previous work on distribution free uniform stability. The authors then apply their results to derive excess risk bounds to inexact $L_0$-ERM.

**Summary Of The Review:**

The problem the paper studied is well related and the authors made substantial contribution for $L_q$-generalization bounds, significantly improving previous results by removing the overhead $\sqrt{N}$ factor. I think the paper is a good contribution to the community and would like to recommend accept.

---

> ### Author Response · Authors · 2022-11-16
> **Response to Reviewer dfYo**
>
> Thank you for your insightful review and appreciation of our work.

---

### Official Review · Reviewer_Hu4j · 2022-11-02

**Confidence:** 3
**Correctness:** 3
**Technical Novelty And Significance:** 3
**Empirical Novelty And Significance:** Not applicable
**Recommendation:** 8

**Clarity, Quality, Novelty And Reproducibility:**

* **Clarity**: The paper is clear and well-written.

* **Quality**: The quality of the paper is good.

* **Novelty**: The gross of the proofs to extend the uniform-stability generalization and excess risk bounds to $L_q$ stability is not novel. However, the results are, as well as the results and proofs of Section 3.

* **Reproducibility**: \
*Theory*: I reproduced all the proofs except for the questions that I placed the authors in the weaknesses. \
*Experiments*: There are no experiments.

**Strength And Weaknesses:**

**Strengths**

* The generalization error bounds under $L_q$-stability are near-optimal, improving upon the current bounds, and matching the rate of their uniform-stability counterparts.
* The excess risk bounds under $L_q$-stability match the rate of the uniform-stability counterparts under the Bernstein condition and add a new form under the quadratic growth condition.
* The bounds on inexact ERM with sparsity constraints include algorithms like iterative hard thresholding (IHT) and match the rate of current bounds under potentially milder conditions.

**Weaknesses**

* There are some parts in the text and the proofs where the contributions and influence of previous work are not clear. It would be good to clarify these parts.

  * The bound in Proposition 1 is *[Celisse and Guedj 2016, Corollary D1]*, which is a Corollary of *[Boucheron et al 2005, Theorem 2]*. Although the text preceding the proposition mentions the references, the wording is slightly confusing.
  * The proof of Theorem 1 essentially follows *[Bousquet et al 2020, proof of Theorem 4]* with little variation to adapt it to $L_q$ stability. The same happens with the proof of Theorem 2 which follows essentially the combination of Lemma 7 and Theorem 4 of *[Bousquet et al 2020]*. Also with the proof of Lemma 5 which follows *[Klochkov and Zhivotovskiy 2021, proof of Lemma 3.1]* and the subsequent beginning and part (a) of the proof of Theorem 3 which follows *[Klochkov and Zhivotovskiy 2021, proof of Theorem 1.1]*.

* In the exposition of the results, sometimes the assumptions are unclear, usually due to the notation $\lesssim$ (which needs to be introduced in the notation section).

  * The standard results from the uniform-stability literature usually require the loss is bounded by some constant, say $L$. This usually appears in (2), (3), (4), and (5), even under the use of $\lesssim$ to make this assumption explicit. Having these constants would help understand better the improvement of using $L_q$-stability. If they are not there, it would be good to explicitly mention these bounds hold under this assumption prior to their introduction.
  * Similarly, in (5) the Bernstein condition constant is usually present to make sure this dependence is clear.
  * The same happens when presenting the contributions in Section 1.2.

    * The generalization bound should have the constant $M_q$ demanding a finite moment $\lVert \ell(A(S);Z) \rVert_q \leq M_q$ or at least an explicit mention that this is required.

    * The excess risk bound should either have the Bernstein constant $B$ and a bounded constant, or an explicit mention that these assumptions are required. Moreover, it should include the Lipschitz constant $G$ and the strong-convexity constant $\mu$  or an explicit mention that these assumptions are required.

* Algorithm 1 seems self-referential. Namely, note that $\tilde{w}_{S,k} := \argmin_{w \in \mathcal{W}, \textnormal{supp}(w) \subseteq \tilde{J}} R_S(w)$ and $\tilde{J} = \textnormal{supp}(\tilde{w}_{S,k})$. This made understanding the section a little more difficult.

* Remark 10 seems a little unfair. It is comparing a bound for misspecified models *[Yuan and Li 2022, Theorem 3]* with a rate $\mathcal{O}(1/N)$ with further assumptions with their $\mathcal{O}(1/\sqrt{N})$ bound on Theorem 4 with weaker assumptions. However, it does not consider the $\mathcal{O}(1/\sqrt{N})$ bound of the same paper *[Yuan and Li 2022, Theorem 1]* which has very similar assumptions (i.e. does not need a bounded parameter space nor a strongly convex population risk as this paper does but requires smoothness of the empirical risk) that can be similarly applicable.

* The paper claims that it uses the developed theory to bound the excess risk of inexact ERM with sparsity constraints. This is not the case, the excess risk bounds are based on $L_q$-stability, yes, but don't use the theorems in Section 2. First, the assumptions are larger in that setting, and second, only some ideas of the proof of Theorem 3 are employed to prove Lemma 6 in the Appendix. \
I believe this should be clarified in the Abstract and the Introduction. Section 3 serves as a motivation for the importance of studying $L_q$-stability, but not for the usage of their generic bounds for $L_q$-stable algorithms, since they are not used there.

* Could you please clarify or write explicitly in the text the final step in the proof of Theorem 3, part (a) and part (b)? That is, the step that follows the "which implies that" and "which then implies".

* The exposition would be clearer if the excess risk bounds and concepts were also introduced in the problem setup instead of that later in Section 2.3.

**References**

*[Boucheron et al 2005]* Moment inequalities for functions of independent random variables. \
*[Celisse and Guedj 2016]* Stability revisited: new generalisation bounds for the leave-one-out. \
*[Bousquet et al 2020]* Sharper bounds for uniformly stable
algorithms. \
*[Klochkov and Zhivotovskiy 2021]* Stability and deviation optimal risk bounds with convergence rate o(1/n). \
*[Yuan and Li 2022]* Stability and risk bounds of iterative hard thresholding.


**Summary Of The Paper:**

This paper continues the recent line of work on high-probability generalization and excess risk bounds for stable algorithms. In this regard the paper:

* Extends the nearly-optimal generalization bounds for uniformly stable algorithms to $L_q$-stable algorithms.
* Similarly, it extends the excess risk bounds for uniformly stable algorithms under the Bernstein condition to $L_q$-stable algorithms under either the Bernstein or quadratic growth conditions.
* It considers the inexact empirical risk minimization (ERM) with sparsity constraints problem. Under Lipschitzness, (high probability) strong convexity, and bounded domain conditions, it shows it is $L_q$-stable and it proves a new excess risk bound for that problem. The bounds are comparable to others in the literature with different assumptions.

**Summary Of The Review:**

This paper extends and builds upon the current best rate bounds on the generalization and excess risk bounds under the uniform-stability assumption to the milder $L_q$-stability assumption.

* For the generalization bounds, instead of further needing a bounded loss, it needs bounded moments.
* For the excess risk bounds, it either maintains the requirement of a bounded loss and the Bernstein condition or requires Lipschitness and the quadratic growth condition. These rates under the latter assumptions are novel as far as I know.

The paper needs to make clearer, though, the influences of the previous literature in their proofs, since some of them follow these papers closely. Similarly, it needs to further clarify some of the assumptions in the introductory text.

Then, the paper finds bounds for the excess risk of inexact ERM with sparsity constraints (which are applicable to IHT). First, they prove these algorithms are $L_q$-stable and then they find specialized bounds that are comparable to those in the literature.

The paper needs to make clearer the comparison of their bounds and assumptions to those in the literature as well as the reason for Section 3, which does not use directly their bounds for generic $L_q$-stable algorithms.

Overall, I believe the paper is well written and is a good contribution, so I recommend acceptance. Nonetheless, I still believe the comments in the weaknesses part should be addressed.

**Minor comments and nitpicks that did not impact the score of the review**

* Please, introduce the notation $\lesssim$.

* In the first paragraph of page 2, "the Efron-Stein inequality".

* In the paragraph before Definition 1, "introduce the ~~following~~ concept of uniform stability*.

* In Remark 1, use either "uniformly bounded" or "uniform boundedness".

* Before (7), please mention explicitly that this holds with probability $1-\delta$.

* Before the acronym HTP, give the name "Hard Threshold Pursuit".

* For completeness, in page 7, after the displayed equation before Definition 3 mentioned that $\tilde{w}_{S,k}$ is the output of Algorithm 1.

* In Section 3, describe what you mean by support to disambiguate with the support of distributions since you are also working with random objects in this work.

* In the last paragraph in Section 4, either "a cardinality constraint" or "cardinality constraints".

* In the Conclusion you don't mention anything about your results on inexact ERM with sparsity constraints. Maybe you want to write a little about that.

* Throughout the text you are using $a$ for the constant on the sub-exponential term and $b$ for the constant on the sub-Gaussian term. However, in the first bullet point of Lemma 4, you reversed them.

* Maybe you want to separate Remark 11 into two remarks.

* At the beginning of Theorem 1 you say "we pad the set with extra...", please write explicitly which set you mean.

* In the last paragraph of page 15: "Based on the triangle and Jensen's inequalities we can show that"

* In (16) I think it should be $4 \sqrt{2 \kappa Nq} M_q$ instead of $2 \sqrt{2 \kappa Nq} M_q$.

* The first inequality of part (a) of Theorem 3 on page 18 and the last inequality of part (b) on page 18 are not due to Hölder's inequality since it only holds for exponents $1 < r < s < \infty$ where the first inequalities are strict. However, it does hold due to Jensen's inequality.

* There is a typo in the first equation in the proof of Lemma 1. In the $\ell(w^* w_{S|J};Z_j)$ it should be $\ell(w^*_{S|J};Z_j)$.

* In (23) instead of inequality $\leq$ it should be $\lesssim$.

* In the array of equations and inequalities after (29) in the second inequality I believe you forgot a factor of $\sqrt{2}$ (it should be a 4 instead of $2 \sqrt{2}$) and it carries over that part.

---

> ### Author Response · Authors · 2022-11-16
> **Response to Reviewer Hu4j**
>
> Thank you for your insightful review and extremely detailed comments for improvement.
>
> > **Your comment:** There are some parts in the text and the proofs where the contributions and influence of previous work are not clear.
>
> **Our response:** Per your comment, in the revised paper we have 1) modified the presentation order of references cited before Proposition 1 to avoid potential confusion, and 2）more explicitly stated in the main text and appendix that our proof techniques for Theorems 1-3 are inspired by those of Bousquet et al. (2020) and Klochkov and Zhivotovskiy (2021) for uniformly stable algorithms, with natural adaptation to the distribution-dependent notion of $L_q$-stability.
>
> > **Your comment:** In the exposition of the results, sometimes the assumptions are unclear, usually due to the notation $\lesssim$ (which needs to be introduced in the notation section).
>
> **Our response:** Many thanks for the suggestions about improving the exposition of results.
>
> 1. We have formally introduced the notation $\lesssim$ in the notation section as requested.
>
> 2. We have improved the exposition of our main results (in Section 1.2) as well as some closely related prior results (in Section 1.1) by explicitly including some key constants in the bounds, such as the bounded-loss, Lipschitz-loss, strong-convexity and Bernstein-condition constatns.
>
> > **Your comment:** Algorithm 1 seems self-referential.
>
> **Our response:** In order to make the statement clearer, we have rephrased the first property of the output $\tilde w_{S,k}$ of Algorithm 1 as: $\tilde w_{S,k}$ is optimal over its support $\tilde J=supp(\tilde w_{S,k})$, i.e., $ \tilde w_{S,k}= \arg\min_{w\in \mathcal{W}, supp(w)\subseteq \tilde J} R_S(w)$.
>
>
> > **Your comment:** Remark 10 seems a little unfair.
>
> **Our response:** Per your comment, we have updated Remark 10 by adding a short discussion on the comparison of our bound to that of Yuan and Li (2022, Theorem 1). Indeed, for misspecified sparsity models, the $\mathcal{O}\big(\frac{1}{\sqrt{N}}\big)$ dominant rate in the sparse excess risk bound of Theorem 4 is comparable to the rate of Yuan and Li (2022, Theorem 1) under similar conditions. Our bound however is more appealing in the sense that it naturally adapts to well-specified models to attain an improved $\mathcal{O}(\frac{1}{N})$ rate in that setting, as discussed in Remark 8. The rate of Yuan and Li (2022, Theorem 1) unfortunately does not have such an ability of adaptation.
>
> > **Your comment:** Section 3 serves as a motivation for the importance of studying $L_q$-stability, but not for the usage of their generic bounds for $L_q$-stable algorithms, since they are not used there.
>
> **Our response:**  Thanks for pointing out this issue which we believe can be addressed by the following clarifications:
>
> 1. It is indeed true that our study of generic $L_q$-stability and generalization theory is largely motivated by the problem of deriving sharper sparse excess risk bounds for IHT-type algorithms which are usually $L_q$-stable rather than uniformaly stable in high-dimensional statistical models.
>
> 2. Technically speaking, the proof of Lemma 7, which is key to the proof of Theorem 4, draws large inspiration from the proof of Theorem 3 yet with some non-trivial adaptation to the sparsity structure of problem.
>
> 3. In the revised manuscript, we have additionally highlighted the motivation purpose of Section 3 and its connection to Section 2 right above the notation section and above the statement of Theorem 4 as well.
>
> > **Your comment:** Could you please clarify or write explicitly in the text the final step in the proof of Theorem 3, part (a) and part (b)? That is, the step that follows the "which implies that" and "which then implies".
>
> **Our response:** Yes, we are more than happy to complete these missing arguments in the revised paper. More specifically, the final steps for obtaining the fast rates in Theorem 3 follow direcly from a simple technical lemma (Lemma 5) about self-bounding inequalities which has been additionally introduced in Appendix A.
>
> > **Your comment:** The exposition would be clearer if the excess risk bounds and concepts were also introduced in the problem setup instead of that later in Section 2.3.
>
> **Our response:** Per your suggestion, we have introduced the concept of excess risk bound at the end of the *Problem setup* section appeared on Page 2. Thanks!
>
> > **Your minor comments**
>
> **Our response:** Many thanks for pointing out these minor issues which have been carefully fixed in the updated draft.
>
> ## References:
> O. Bousquet, Y. Klochkov, and N. Zhivotovskiy. Sharper bounds for uniformly stable
> algorithms. *COLT*, pp. 610–626, 2020.
>
> Y. Klochkov and N. Zhivotovskiy. Stability and deviation optimal risk bounds with convergence rate O(1/n). *Advances in NeurIPS*, 34:5065–5076, 2021.
>
> X.-T. Yuan and P. Li. Stability and risk bounds of iterative hard thresholding. *IEEE Trans. on Information Theory*, 2022.

---

> > ### Comment · Reviewer_Hu4j · 2022-11-21
> > **Answer to response**
> >
> > Thank you for your response. I see most of the comments have been satisfactorily addressed and the paper reads better overall now. Just one further comment.
> >
> > * *Regarding the prior work*:
> >
> > In the revised version I see how the influence from the work *[Bousquet et al. 2020]* is already clarified for Theorems 1, 2, and 4. However, I don't see how the influence of *[Klochkov and Zhivotovskiy 2021]* is clarified for Lemma 5 (new Lemma 6).

---

> > > ### Author Response · Authors · 2022-11-21
> > > **Appreciate your acknowledgment and additional comment**
> > >
> > > Dear Reviewer Hu4j
> > >
> > > We are very glad to know that our response can address your concerns raised in the previous review. Many thanks  for providing an additional comment regarding the influence of prior work. We would absolutely like to highlight in the next draft that the new Lemma 6 is a strict analogue of the result by Klochkov and Zhivotovskiy (2021, Lemma 3.1) under the distribution-dependent notion of $L_q$-stability.
> > >
> > > With best regards,
> > >
> > > Authors of Paper2865

---

### Author Response · Authors · 2022-11-16
**General response**

We sincerely thank all the reviewers for their insightful review of our work and possitive feedbacks. We have carefully revised the manuscript based on the comments which are very constructive for improvement. The following is a summary of major changes:

1. We have further clarified, both in the main paper and appendix sections, that our proof techniques for the Theorems in Section 2 follow closely those of Bousquet et al. (2020) and Klochkov and Zhivotovskiy (2021) for uniformly stable algorithms, with natural generalization to the weaker notion of $L_q$-stability.

2. In the introduction section, the exposition of our main results as well as some closely related prior results has been improved by explicitly showing some key factors (such as the bounded-loss, Lipschitz-loss, strong-convexity and Bernstein-condition constatns) involved in the bounds.

3. In Remark 10, the comparison with the sparse excess risk bonds of Yuan and Li (2022) for IHT has been expanded.

4. In Remark 6, we have updated the discussions regarding the bounded-loss conditions in Theorem 3.

5. The connection of the proof technqies of Theorem 4 to those of Theorem 3 has been highlighted in the relevant context.

6. We have fixed the typos and other minor issues mentioned in the reviews.

The major changes are highlighted in red text. We sincerely hope that the given concerns have been addressed satisfactorily in the revised manuscript and the point-by-point responses to the reviewers' comments.

---

> ### Comment · Area_Chair_vfES · 2022-12-13
> **additional concerns**
>
> I've noticed some additional issues with this work:
> The significance of this results does not appear to be well justified. Indeed, while prior work has studied $L_q$ stability, so far no applications of  this notion (beyond $L_2$ and uniform stability covered in existing work) that do not follow, with a bit of care from uniform stabillity have been shown.
> In particular, the application suggests that $L_q$ stability is used in a crucial way. However it appears that it is used only to deal with the low-probability of failure given in assumption 4. There are other easy way to deal with such low probability events (for example by truncating the random variables appropriately). Such low probability failure events can usually be relatively easily incorporated into concentration inequality. A clear discussion of why such simpler approaches would not suffice seems appropriate to evaluate the application. Without a notable new application or new technical ideas this work would be more suitable for a journal.
>
> I've also noticed a bibliographic issue. The abstract and intro attribute the strong generalization bounds for uniformly stable algorithms to Bousquet et al, 2020. However the result are due to Feldman and Vondrak (2019) (as can be immediately seen from the abstract of Bousquet et al 2020). Feldman and Vondrak removed the $\sqrt{n}$ factor from the known result. Bousquet et al. sharpened the result by improving the constant factor and a second order term. Their proof technique builds heavily on Feldman and Vondrak 2019.

---

> > ### Author Response · Authors · 2022-12-13
> > **Response to Area Chair vfES**
> >
> > Dear Area Chair vfES,
> >
> > We appreciate very much your great time and efforts dedicated in shepherding the review of our paper. Since the additional comments were received at the last minite of paper discussion phase, we had rather limited opportunity to respond. We would be happy to make further and more detailed clarifications if required and allowed.
> >
> > The main concern revolves around the value of our $L_q$-stability theory added beyond the classic uniform stability theory (and its variants), which we sincerely hope can be addressed by the following brief clarifications:
> >
> > 1. Concerning the addressed case where the algorithmic stability holds with high probability (over data), we agree that the traditional McDiarmid’s inequality implied exponential generalization bounds under uniform stability can be more or less straightforwardly extended to such a non-uniform regime. (See the related work briefly surveyed in the second paragraph of Section 4). However, when it comes to the recent break-through bounds of Feldman & Vondrak (2019); Bousquet et al. (2020), it is substantially less obvious to us how to *easily* extend these near-optimal bounds under the almost-everywhere stability via simply *incorporating the low probability failure events into concentration inequality*. This is actually in sharp contrast to what have been done by Feldman & Vondrak (2019, Theorem 4.5); Bassily et al. (2020, Theorem 2.1) for stochastic optimization algorithms when the uniform stability holds with high probability over the internal randomness of algorithm, rather than the randomness of data.
> >
> > 2. Continuing with the above reply, it is worthwhile to highlight that a fundamental principle of this work is trying to develop a sharper yet simple generalization theory under the generic notion of $L_q$-stability to address those algorithms having ``uniform stability with high probability'' rather than being uniformly stable over data distribution. While it is definitely possible to figure out other alternative approaches to achieve the similar goal, we have the feeling that our present $L_q$-stability arguments are already powerful and elegant enough to do the job.
> >
> > 3. Regarding the novelty of application, we believe that the application of our sharper $L_q$-stability bounds to the sparse excess risk analysis of inexact $L_0$-ERM is novel and concrete in the following senses:
> >
> >    (1) it answers a call by Celisse and Guedj (2016) for extending the range of applicability of the $L_q$-stability thoery beyond unbounded ridge regression, and is clearly different from other representative applications of $L_q$-stabiity theory including $k$-nearest neighbor classification and $k$-folds cross-validation (Celisse & Mary-Huard, 2018; Abou-Moustafa & Szepesvari, 2019);
> >
> >    (2) technically speaking, the proof of Lemma 7, which is key to the proof of Theorem 4, is highly non-trivial with several new ingredients developed for handing the challenges introduced by the combinatorial optimization nature of $L_q$-ERM;
> >
> >    (3) the obtained sparse excess risk bounds in Theorem 4 are new and more adaptive the sparsity structure of models, yet under milder conditions than the SOTA analysis (Yuan & Li, 2022);
> >
> >    (4) Last but not least, it is expected that our theory can be readily applied to the above mentioned prior applications to obtain sharper generalization bounds.
> >
> >
> > With regard to the bibliographic issue, we would like to clarify that we indeed have mentioned at the top line of Page 2 that the best known exponential generalization bounds for uniformly stable algorithms are offered by Feldman & Vondrak (2019); Bousquet et al.(2020). In our work, we choose to follow the techniques of Bousquet et al.(2020) since their bounds are slightly sharper while the arguments are relatively simpler. Per your comment, we plan to further highlight the connections of our results to those of Feldman & Vondrak (2019) in the updated draft. Thanks!
> >
> >
> > With best regards,
> >
> > Authors of Paper2865
> >
> > ## References:
> >
> > V. Feldman and J. Vondrak. High probability generalization bounds for uniformly stable algorithms with nearly optimal rate. In *COLT*, 2019.
> >
> > R. Bassily, V. Feldman, C. Guzman, and K. Talwar. Stability of stochastic gradient
> > descent on nonsmooth convex losses. In *NeurIPS*, 2020.
> >
> > O. Bousquet, Y. Klochkov, and N. Zhivotovskiy. Sharper bounds for uniformly stable
> > algorithms. In *COLT*,2020.
> >
> > A. Celisse and B. Guedj. Stability revisited: new generalisation bounds for the leave-oneout. *arXiv:1608.06412*, 2016.
> >
> > A. Celisse and T. Mary-Huard. Theoretical analysis of cross-validation for estimating the
> > risk of the k-nearest neighbor classifier. *JMLR*, 2018.
> >
> > K. Abou-Moustafa and C. Szepesvari. An exponential efron-stein inequality for ´ $l_q$ stable
> > learning rules. In *ALT*, 2019.
> >
> > X.-T. Yuan and P. Li. Stability and risk bounds of iterative hard thresholding. *IEEE Trans. Information Theory*, 2022.

---

> > > ### Comment · Area_Chair_vfES · 2022-12-14
> > > **applications matter**
> > >
> > > Thank you for the response. I agree that such general bounds are of interest to experts. My point is that how interesting the bounds are for the broader community depends heavily on whether they lead to some generalization results beyond the reach of existing techniques. A clear explanation why the given application cannot be obtained via simple adaptation of existing techniques is not given in the submission itself. This is a rather glaring omission since previous attempts at deriving substantially new generalization bounds via $L_q$ stability have not been successful and there are a number of simple approaches for dealing with very low probability events.

---

### Decision · Program_Chairs · 2023-01-20

**Decision:**

Accept: poster

**Justification For Why Not Higher Score:**

The included application is very specialized and even for that application the paper does not explain why $L_q$ stability is actually necessary. So my sense is that its appeal is going to be rather narrow at ICLR.

**Justification For Why Not Lower Score:**

This is solid theoretical work that could turn out to be useful for deriving more interesting generalization bounds.

**Metareview: Summary, Strengths And Weaknesses:**

This work extends recently derived nearly optimal generalization bounds based on uniform stability to a weaker notion of $L_q$ stability. An application of the new bound to a certain setting of sparse estimation is given. This is an interesting generalization that allows one to apply strong stability based generalization bounds in more general contexts.  The techniques are a relatively direct extension of existing proofs with appropriate assumptions on $L_q$ stability and boundedness of the loss in $q$ norm used in place of uniform bounds. The application is rather specialized but also appears to be novel. A more detailed discussion of the necessity and other potential applications of $L_q$ stability would greatly benefit this work.

**Note From Pc:**

if the above contains the word "oral" or "spotlight" please see: "oral" presentation means -> notable-top-5% and "spotlight" means -> notable-top-25%. As stated in our emails, we are disassociating presentation type from AC recommendations